# SpatialBoost: Enhancing Visual Representation through Language-Guided Reasoning

## Abstract

Despite the remarkable success of large-scale pre-trained image representation models (i.e., vision encoders) across various vision tasks, they often fail to learn 3D spatial relationships between objects and backgrounds in the real world, constraining their effectiveness in various downstream applications. We attribute this to the limited availability of large-scale 3D training data, which makes it difficult for current image representation learning approaches to learn spatial relationships. This motivates the need for learning paradigms that rely on strong supervision while requiring less data. To address this, we propose a novel learning framework that enhances the spatial awareness of existing pre-trained vision encoders by injecting dense 3D spatial knowledge expressed in linguistic forms. To be specific, the core idea involves converting dense 3D spatial information from 2D images into linguistic expressions, which is then used to inject such spatial knowledge into vision encoders through a Large Language Model (LLM). To this end, we adopt a multi-turn Chain-of-Thought (CoT) reasoning process that progressively incorporates dense spatial knowledge and builds hierarchical spatial understanding. To validate effectiveness, we adapt SpatialBoost to state-of-the-art vision encoders such as DINOv3, and evaluate its performance gains on a wide range of benchmarks requiring both 3D perception and general vision abilities.

## 1 Introduction

Pre-trained image representation models (He et al., 2020; Donahue & Simonyan, 2019; Chen et al., 2020b; Dosovitskiy et al., 2021; Li et al., 2023b; Assran et al., 2023) have shown remarkable success in various downstream tasks, such as image classification (Krizhevsky et al., 2009; Cui et al., 2018), semantic segmentation (Lin et al., 2014; Zhou et al., 2019), monocular depth prediction (Silberman et al., 2012; Geiger et al., 2012), and vision-language understanding (Antol et al., 2015; Hudson & Manning, 2019). The core idea behind these successes is extracting transferrable representation from large-scale image datasets such as ImageNet (Deng et al., 2009), enabling the model to understand semantic information within images that is significantly useful for various downstream tasks.

Despite their success, these models are predominantly trained on 2D images and hence face a fundamental challenge in acquiring 3D spatial awareness capabilities. Consequently, large vision language models struggle to discern 3D spatial relationships between objects in images (Liu et al., 2023a; Fu et al., 2024b; Wang et al., 2025b; Cheng et al., 2024), and demonstrate sub-optimal performance in vision-based robotic control tasks compared to approaches that directly utilize 3D information (Ze et al., 2024; Ke et al., 2024; Zhen et al., 2024). To address these limitations, several works train vision models on multi-view images that naturally encode spatial information (Zhang et al., 2024; Wang et al., 2024b; Charatan et al., 2024). While these approaches have shown promise in robot control tasks (Seo et al., 2023; Sermanet et al., 2018), their broader applicability remains constrained by the need to use carefully curated data (Yu et al., 2023) or obtain multi-view datasets from simulation environments (Savva et al., 2019), creating significant limitations for scaling up these approaches. These challenges highlight the need for a novel framework that enables effective learning of 3D information with substantially less data.

However, we note that vision models specialized for individual tasks are able to infer object positions and point depths from standard 2D images. These extracted cues make it possible to extend spatial information by modeling geometric relationships between objects in a scene. We hypothesize that

Figure 1: **Overview of SpatialBoost.** We enhance spatial and geometric understanding of pre-trained vision encoders by leveraging language-guided spatial reasoning. SpatialBoost consists of (a) spatial knowledge extraction through depth estimation, 3D reconstruction, segmentation, and region captioning, (b) converting spatial knowledge into multi-turn spatial reasoning from pixel to scene levels, and (c) building a spatial-aware vision encoder with LLM using generated data in (b).

such spatial information can be systematically converted into explicit representations by leveraging language. Moreover, since language naturally composes information in a sequential and structured form, this property allows the construction of labels that capture dense spatial relationships within a scene.

Based on these insights, we introduce SpatialBoost, a training framework that enhances the spatial understanding of pre-trained vision encoders by leveraging language-guided reasoning (see Figure 1). We inject linguistically described spatial knowledge through decoder-based fine-tuning with Large Language Models (LLM), where the model takes single or multi-view images as input and generates descriptions. In particular, to leverage this knowledge without forgetting the existing knowledge, we incorporate additional learnable parameters (*i.e.*, dual-channel attention module) into the vision encoder and train only them while freezing the existing parameters. Furthermore, to incorporate dense spatial information in a structured manner, we present a multi-turn visual spatial reasoning approach that builds hierarchical spatial understanding through pixel-level, object-level, and scene-level sub-questions and answers.

To validate the effectiveness of our method, we apply SpatialBoost to state-of-the-art image encoders, including DINOv3 (Siméoni et al., 2025) and SigLIPv2 (Tschannen et al., 2025), and evaluate them across a diverse set of vision tasks: monocular depth estimation, semantic segmentation, 3D scene understanding, vision-based robotic control, image classification, image retrieval, spatial reasoning, and general VQA.[1] Our experiment first shows that SpatialBoost consistently improves performance on tasks requiring 3D spatial knowledge. For example, on the 3D scene understanding task, SpatialBoost improves DINOv3 by 3.5% (51.4% → 54.9%) on the SQA3D task from Lexicon3D Benchmark (Man et al., 2024). In addition, on depth estimation tasks, SpatialBoost improves SigLIPv2 from an RMSE score of 0.51 to 0.39 on NYUd linear probing. Moreover, we show that SpatialBoost even improves the performance of the vision encoders across all benchmarks, notably in image classification: SpatialBoost improves ImageNet linear probing performance of DINOv3 from 88.4% to 90.2%.

## 2 RELATED WORK

**Self-supervised Learning for Image Representation.** In earlier years, most approaches relied on supervised learning with large-scale labeled datasets to train models (Deng et al., 2009; Simonyan & Zisserman, 2014; Szegedy et al., 2014; He et al., 2016). However, the dependence on annotated data introduced scalability challenges due to label expense. To address this, self-supervised learning (SSL) has emerged as a dominant paradigm, leveraging unlabeled data to learn image representations. Contrastive learning methods, including SimCLRv2 (Chen et al., 2020c), MoCov3 (Chen et al., 2021), DINOv2 (Oquab et al., 2023), and iBOT (Zhou et al., 2021), are trained to distinguish between representations of augmented views of the same image and those of different images. Concurrently, mask prediction approaches such as BEiT (Bao et al., 2021) and MAE (He et al., 2022),

---

[1]Due to space constraints, results on spatial reasoning and general VQA tasks are provided in the appendix.

learn representations by reconstructing masked portions of input images. While these methods excel at capturing rich semantic features within 2D images, they lack mechanisms to effectively encode 3D spatial knowledge. On the other hand, we overcome this limitation by enhancing image representations through a novel method that injects 3D spatial knowledge by utilizing language decoding.

**Multi-modal Learning for Image Representation.** The increasing prominence of multi-modal tasks has catalyzed the development of vision-language models that jointly represent visual and textual information. These models typically employ weakly supervised learning by leveraging text caption. Contrastive learning schemes, *e.g.*, CLIP (Radford et al., 2021), SigLIP (Zhai et al., 2023) and OpenCLIP (Cherti et al., 2023), consist of vision and text encoders and are trained to align their representations in a shared embedding space. Alternative methodologies like M3AE (Geng et al., 2022), jointly encode image patches and text tokens, employing masked prediction objectives to reconstruct both modalities. More recently, autoregressive formulations such as iGPT (Chen et al., 2020b), have emerged, treating image patches and text tokens as sequential elements for predictive modeling. These approaches successfully enrich visial representations with semantic context derived from natural language descriptions. However, existing models necessitate joint pre-training of both modalities from scratch, imposing significant computational demands and preventing efficient adaptation of existing pre-trained models. Our method eliminates the need for joint text-image representation learning by using LLM, thereby enhancing pre-trained models with relevant linguistic information efficiently.

**Multi-View Learning for Image Representation.** Recent advances in vision tasks that require 3D spatial understanding and generation have increased the demand for effective 3D spatial representations (Chen et al., 2024b; Wu et al., 2024; Goyal et al., 2023; Shridhar et al., 2023). Multi-view images from different camera viewpoints or video sequences serve as input for these tasks. Our focus is specifically on augmenting image representations with useful 3D information. Typically, following approaches similar to single-view image representation learning, multi-view data has been processed by converting images into patches for masked prediction such as MV-MWM (Seo et al., 2023) or through contrastive learning methods (Sermanet et al., 2018). Additionally, to learn 3D-related information more explicitly, approaches that predict 3D features from image representation (Ke et al., 2024; Gervet et al., 2023; Ze et al., 2024) have been proposed. These approaches have led to significant performance improvements in vision-based robot control. However, such methods are limited by multi-view data, making it difficult to develop them into pre-trained models for general 3D understanding. Our approach proposes a method to learn 3D spatial representations from both single-view and multi-view images, avoiding these limitations.

# 3 METHOD

In this section, we introduce SpatialBoost, a visual representation learning framework designed to improve vision encoders by injecting 3D spatial information expressed in natural language. We first present a multi-modal architecture that incorporates linguistically expressed visual information into the vision encoder through a dual-channel attention layer, ensuring that original visual features are preserved while 3D spatial information is fully exploited (see Section 3.1). On top of this architecture, we design a Visual-Question-Answering (VQA) dataset that hierarchically disentangles 3D spatial relations from both single/multi-view images, enabling the vision encoder to learn spatial information more effectively (see Figure 1).

## 3.1 TRAINING PIPELINE

To train a vision encoder from rich spatial information encoded in large-scale linguistic expressions, our key idea is to utilize Large-Language Models (LLM) by constructing a multi-modal architecture composed of a vision encoder $f_V$, a trainable projection module $g_P$, and the LLM $f_L$. However, without proper alignment between visual and textual representations, the training signals from the LLM cannot effectively propagate back to the vision encoder, making the learning process ineffective. To fully exploit language supervision, we begin by aligning the visual encoder with the textual embedding space of the LLM. Specifically, we adopt LLaVA (Liu et al., 2023b), a two-stage training for the alignment: feature alignment (Stage 1) and visual instruction tuning (Stage 2). After the alignment, we introduce a training framework that uses a language-guided reasoning dataset to

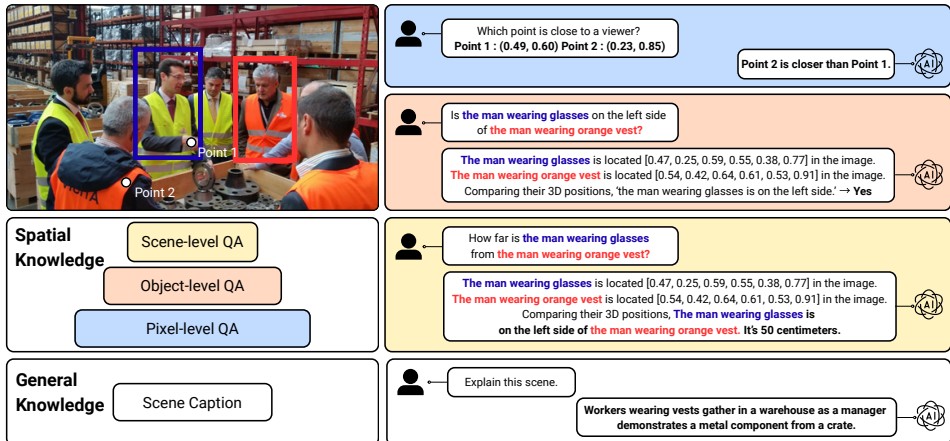

Figure 2: **Illustration of multi-turn visual spatial reasoning dataset**, exhibiting pixel-level, object-level, and scene-level reasoning QAs. At the pixel-level, the QA task queries the 3D positions of points (*e.g.*, via depth estimation). At the object-level, it extracts spatial properties of objects (*e.g.*, by predicting bounding cubes or relative positions). At the scene-level, it determines the exact distances between multiple objects that require the rationales of the previous steps. At last, we add 2-turn for general scene caption. These are listed in order and constitute 12 multi-turn visual spatial reasoning conservation.

fine-tune the vision encoder (Stage 3). Notably, direct full fine-tuning in this final stage would lead to catastrophic forgetting of the pre-trained knowledge embedded in the vision encoder. To address this challenge, we introduce *dual-channel attention* layers that enable the model to acquire spatial understanding while preserving its original representational capabilities.

Formally, given an input image $\mathbf{x}$ and multi-turn conversation data $(\mathbf{x}_q^1, \mathbf{x}_a^1, \cdots, \mathbf{x}_q^T, \mathbf{x}_a^T)$ from question-answering (QA) pairs $(Q_\mathbf{x}, A_\mathbf{x})$, we first encode $\mathbf{x}$ to obtain visual features $\mathbf{z}_\mathbf{v} = f_V(\mathbf{x})$, which are mapped into the token embedding space via $g_P(\mathbf{z}_\mathbf{v})$. These visual tokens are then concatenated with text tokens and fed into the LLM. Given the multi-turn conversation data and input image, we optimize the model through autoregressive loss. Our training pipeline consists of three stages and all stages are trained with supervised fine-tuning (SFT) loss. We describe each stage in the following paragraphs.

**Stage 1: Feature alignment.** In this stage, we train a projector $g_P$ that maps image features into the textual embedding space of the LLM. This projector pre-training contributes to the stable vision-language alignment. Following the training setup in multi-modal large language models (Liu et al., 2023a; 2024a), we freeze the parameters of both the visual encoder $f_V$ and the language model $f_L$, and optimize only the projector $g_P$.

**Stage 2: Visual instruction tuning.** Following the projector alignment in Stage 1, this stage extends the alignment to the LLM. We freeze the visual encoder $f_V$ and fine-tune the projector $g_P$ and the language model $f_L$ using our multi-view VQA data, combined with the single-view visual instruction data from LLaVA (Liu et al., 2023a). This step enables $f_L$ and $g_P$ to handle multi-view visual questions. We provide details of proposed multi-view VQA data in Section 3.2.

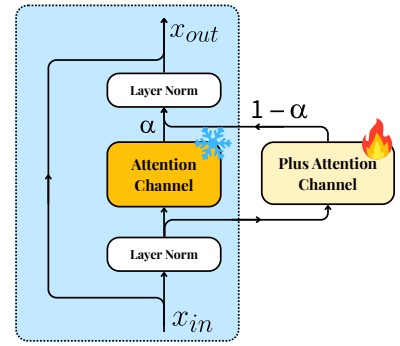

Figure 3: **Illustration of the dual-channel attention layer** (Hong et al., 2023a), where an additional attention block is introduced alongside the original attention block and merged via a learnable mixture factor $\alpha$.

**Stage 3: Vision encoder fine-tuning with dual-channel attention.** Finally, we fine-tune the vision encoder $f_V$ to have the capability of spatial understanding. To effectively inject dense spatial knowledge into the vision encoder, we use multi-turn visual spatial reasoning dataset (see Section 3.2), which is carefully designed for hierarchical spatial reasoning. We train the vision encoder $f_V$ and

Table 1: **Results on monocular depth estimation** from NYUd (Silberman et al., 2012) and KITTI (Geiger et al., 2013) benchmarks. We report the RMSE score between ground truth and predicted depth values. Lower is better. For all results, we freeze the encoder backbone and train a linear head (lin.) or DPT head (Ranftl et al., 2021) on top of the image features of the last layer.

Table 2: **Results on semantic segmentation** from ADE20K (Zhou et al., 2017) and Pascal VOC (Everingham et al., 2010) benchmarks. We report mIoU score. Higher is better. For all results, we freeze the encoder backbone and report results of linear probing (lin.) or multi-scale evaluation (+ms), where the multi-scale approach uses features from the last four layers of the visual encoder to perform segmentation.

| | NYUd | | KITTI | |
|---|---|---|---|---|
| Method | lin. | DPT | lin. | DPT |
| OpenCLIP | 0.53 | 0.41 | 3.54 | 2.70 |
| +SpatialBoost (Ours) | **0.40** | **0.38** | **2.79** | **2.54** |
| SigLIPv2 | 0.51 | 0.40 | 3.32 | 2.64 |
| +SpatialBoost (Ours) | **0.39** | **0.34** | **2.71** | **2.50** |
| DINOv2 | 0.37 | 0.29 | 2.60 | 2.11 |
| +SpatialBoost (Ours) | **0.30** | **0.25** | **2.53** | **2.07** |
| DINOv3 | 0.31 | 0.25 | 2.33 | 2.02 |
| +SpatialBoost (Ours) | **0.25** | **0.21** | **2.20** | **1.84** |

| | ADE20K | | Pascal VOC | |
|---|---|---|---|---|
| Method | lin. | +ms | lin. | +ms |
| OpenCLIP | 39.5 | 46.0 | 71.7 | 79.3 |
| +SpatialBoost (Ours) | **40.5** | **47.3** | **75.1** | **80.9** |
| SigLIPv2 | 42.8 | 48.7 | 72.6 | 79.1 |
| +SpatialBoost (Ours) | **45.1** | **50.8** | **79.0** | **82.2** |
| DINOv2 | 49.3 | 53.0 | 83.0 | 86.2 |
| +SpatialBoost (Ours) | **52.0** | **54.9** | **84.5** | **87.6** |
| DINOv3 | 55.9 | 60.3 | 86.6 | 89.8 |
| +SpatialBoost (Ours) | **59.7** | **63.1** | **88.5** | **90.9** |

the projection module $g_P$ while keeping the parameters of the LLM $f_L$ frozen, allowing only the vision encoder to benefit from language-driven spatial information. We employ SFT loss, and through this training process, the vision encoder learns to extract meaningful representations necessary for producing answers. However, direct full fine-tuning risks forgetting of the pre-trained knowledge embedded in the vision encoder. To address this challenge, we introduce a dual-channel attention mechanism (see Figure 3). Specifically, for each attention layer $\text{Attn}(\cdot)$ in the visual encoder $f_V$, we introduce an additional attention layer $\text{Attn}^+(\cdot)$, whose weight parameters are initialized to the same values as those of $\text{Attn}(\cdot)$. Given an input $\mathbf{x}$ to each attention layer, we merge the outputs of $\text{Attn}(\cdot)$ and $\text{Attn}^+(\cdot)$ by introducing a trainable mixture factor $\boldsymbol{\alpha} = \text{sigmoid}(\mathbf{a}) \in (0,1)^d$ with zero-initialized parameter $\mathbf{a} \in \mathbb{R}^d$, where $d$ is the hidden dimension of $\mathbf{x}$, as follows:

$$\text{Attn}^{\text{final}}(\mathbf{x}) = \boldsymbol{\alpha} \cdot \text{Attn}(\mathbf{x}) + (1 - \boldsymbol{\alpha}) \cdot \text{Attn}^+(\mathbf{x}). \tag{1}$$

During fine-tuning, we only update the parameters of $\text{Attn}^+$ and $\boldsymbol{\alpha}$ while keeping all other parameters frozen. This approach allows the vision encoder to initially rely on pre-trained attention weights and gradually incorporate new attention weights, smoothly enhancing spatial awareness without discarding existing knowledge (see classification result in Figure 6).

## 3.2 ENHANCING VISION ENCODER WITH SPATIAL CoT

To effectively inject dense spatial information into vision encoders, we address the fundamental limitations of existing spatial datasets. Current spatial VQA data consist of simple single-turn QA pairs with limited information content, insufficient for transferring comprehensive 3D understanding. To overcome this limitation, We introduce Multi-view VQA, which helps align the vision encoder with the LLM to effectively handle multi-view data and a multi-turn Chain-of-Thought (CoT) framework (Wei et al., 2022) for both single-view and multi-view images that enables the injection of substantially richer spatial information in a single training instance.

**Multi-view VQA Dataset.** To enhance multi-view VQA capabilities during the visual instruction tuning (Stage 2), we construct multi-view VQA dataset. We first apply LPIPS (Zhang et al., 2018) metric to the 3D or video dataset to obtain a pair of images. Given the pair of images, we employ GPT-4o (Achiam et al., 2023) to generate visual questions targeting general multi-view knowledge. We provide more details in Section C.

**Multi-Turn Visual Spatial Reasoning Dataset.** To enhance spatial reasoning capabilities of the vision encoder (Stage 3), we construct multi-turn visual spatial reasoning dataset for single-view and multi-view. Additionally, to enhance general knowledge of the vision encoder, we append GPT-generated scene captions after spatial reasoning turn. For single-view image, we first extract a 3D point cloud from given an image $\mathbf{x}$ by applying diverse vision models (*e.g.*, depth estimation

Table 3: **Results on 3D-centric tasks.** We evaluate unified probing on diverse 3D-related tasks from ScanNet (Dai et al., 2017) scenes. We report BLEU-1 score for Vision-Language Reasoning (VLR) on ScanQA (Azuma et al., 2022) and SQA3D (Ma et al., 2023). For Visual Grounding (VG), we report accuracy on overall category of ScanRefer (Chen et al., 2020a) dataset. For Geometric Understanding (GU), we report Registration Recall (RR) at 0.05m RMSE threshold and Relative Translation Error (RTE). For 3D Semantic Understanding (3D SU), we report accuracy and mIoU. Lower is better for RTE and higher is better for all other metrics.

| | VLR | | VG | GU | | 3D SU | |
|---|---|---|---|---|---|---|---|
| Method | ScanQA ↑ | SQA3D ↑ | ScanRefer-Overall ↑ | RR@0.05m (%) ↑ | RTE (m) ↓ | Acc ↑ | mIoU ↑ |
| OpenCLIP | 36.9 | 48.0 | 50.1 | 22.6 | 0.40 | 39.8 | 6.9 |
| +SpatialBoost (Ours) | **39.2** | **49.9** | **56.6** | **78.8** | **0.17** | **76.9** | **54.9** |
| SigLIPv2 | 38.1 | 48.5 | 51.4 | 47.8 | 0.28 | 47.7 | 9.2 |
| +SpatialBoost (Ours) | **40.8** | **50.1** | **56.8** | **86.4** | **0.15** | **81.0** | **55.5** |
| DINOv2 | 39.5 | 49.8 | 52.7 | 82.4 | 0.15 | 83.0 | 64.1 |
| +SpatialBoost (Ours) | **40.3** | **50.4** | **57.0** | **92.4** | **0.13** | **89.8** | **68.3** |
| DINOv3 | 40.6 | 51.4 | 56.2 | 86.9 | 0.10 | 91.1 | 69.1 |
| +SpatialBoost (Ours) | **43.3** | **54.9** | **61.1** | **97.5** | **0.06** | **91.9** | **70.6** |

model (Bochkovskii et al., 2024) and image segmentation model (Ravi et al., 2024)). For multi-view images $\{\mathbf{x}_1, \cdots, \mathbf{x}_N\}$, we use 3D reconstruction model (Wang et al., 2025a) to extract a 3D point cloud from given images. Using the point cloud, we synthesize QA pairs specialized in spatial reasoning about $\mathbf{x}$ or $\{\mathbf{x}_1, \cdots, \mathbf{x}_N\}$.

We then design spatial reasoning QA pairs at three hierarchical levels: pixel, object, and scene, enabling LLM to perform CoT reasoning from narrow to broad view. Specifically, at the pixel-level, the QA task is designed to capture the overall geometry in the image by querying the absolute or relative 3D position of a point, *e.g.*, "What is the depth value at coordinate $(x, y)$?". At the object-level, the QA task tackles the semantic spatial information of objects inside the image using a bounding cube of the object in 3D space, *e.g.*, "Is [A] on the left side of [B]?", where [A] and [B] is the descriptions about the object in image. We note that this level uses the pixel-level spatial information as a rationale, enabling LLM to reason about the geometry of objects in 3D space. Lastly, at the scene-level, the QA task is designed to predict the exact distance between multiple objects that requires coherent 3D spatial understanding, *e.g.*, "How far is [A] from [B]?".

## 4 EXPERIMENTS

Through extensive experiments, we validate the performance of SpatialBoost and ablate its key components, focusing on following questions:

- Can SpatialBoost improve spatial knowledge of the vision encoder? (Tables 1 to 4)
- Isn't SpatialBoost overfitted to spatial knowledge? (Table 5)
- Which components contribute to SpatialBoost performance? (Table 6 and Figure 6)

### 4.1 EXPERIMENTAL SETUP

**VQA Dataset Construction.** For single-view image, we use randomly sampled 100K images from the SA1B dataset (Kirillov et al., 2023) to construct the single-view VQA dataset specialized in chain-of-thought spatial reasoning. For multi-view images, we use filtered 200K samples from the ego-centric video dataset (Grauman et al., 2022) and 3D dataset (Jensen et al., 2014; Dai et al., 2017; Mildenhall et al., 2021; Barron et al., 2022) to construct multi-view VQA dataset niche in multi-view reasoning or alignment. More details in Section D.

**Baselines.** For all experiments, we compare our methods with the recent widely-used pre-trained image representation models. To be specific, we first consider OpenCLIP (Cherti et al., 2023) ViT-G/14 and SigLIPv2 (Tschannen et al., 2025) ViT-g/16, known for language-aligned vision encoder. We also consider DINOv2 (Oquab et al., 2023) ViT-g/14 and DINOv3 (Siméoni et al., 2025) ViT-7B/16, which is a recent state-of-the-art vision encoder.

Table 4: **Results on vision-based robot learning.** We report the performance of imitation learning agents on 4 domains from CortexBench (Majumdar et al., 2023), which are trained upon the image representations. In particular, we report the normalized score for DMControl and success rates (%) for other tasks.

| Method | Adroit | MetaWorld | DMControl | Trifinger | Avg. |
|---|---|---|---|---|---|
| OpenCLIP | 52.6 ± 4.9 | 83.0 ± 2.7 | 58.5 ± 1.9 | 67.7 ± 0.5 | 65.5 |
| +SpatialBoost (Ours) | **61.1** ± 3.4 | **87.0** ± 3.3 | **61.0** ± 1.6 | **72.9** ± 0.3 | **70.5** |
| SigLIPv2 | 56.5 ± 3.0 | 84.7 ± 2.9 | 69.4 ± 2.1 | 68.3 ± 0.8 | 69.7 |
| +SpatialBoost (Ours) | **66.5** ± 1.9 | **89.1** ± 0.9 | **73.5** ± 1.8 | **73.9** ± 0.7 | **75.8** |
| DINOv2 | 55.4 ± 2.7 | 82.4 ± 4.0 | 67.9 ± 1.0 | 66.8 ± 0.2 | 68.1 |
| +SpatialBoost (Ours) | **68.1** ± 2.9 | **88.5** ± 3.1 | **75.0** ± 1.1 | **71.4** ± 0.8 | **75.8** |
| DINOv3 | 63.9 ± 1.5 | 83.8 ± 1.6 | 70.8 ± 1.8 | 72.8 ± 0.5 | 72.8 |
| +SpatialBoost (Ours) | **71.8** ± 3.4 | **92.0** ± 1.9 | **80.4** ± 2.4 | **79.0** ± 0.6 | **80.8** |

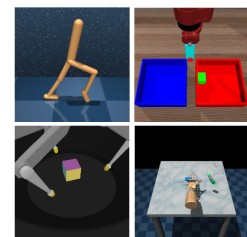

Figure 4: **Examples of visual observations from CortexBench.** We train imitation learning agents to learn a mapping from these visual observations to expert actions.

**Implementation Details.** We choose Qwen-2.0-7B (Yang et al., 2024) as the LLM backbone and 2-layer MLP as the projector, following the architecture of LLaVA-1.5 (Liu et al., 2024a). Further details are provided in Section A.

### 4.2 DENSE PREDICTION TASKS

**Setup.** We evaluate SpatialBoost on dense prediction tasks requiring geometric and semantic spatial understanding. For geometric understanding, we perform monocular depth estimation on NYUd (Silberman et al., 2012) and KITTI (Geiger et al., 2013) using linear or DPT (Ranftl et al., 2021) heads. For semantic understanding, we evaluate on ADE20K (Zhou et al., 2017) and Pascal VOC (Everingham et al., 2010) segmentation benchmarks using linear or multi-scale heads. All experiments freeze the visual backbone during training (see Section A for details).

**Results.** As shown in Table 1 and 2, SpatialBoost consistently improves both geometric and semantic spatial understanding across various encoders. For instance, OpenCLIP's RMSE on NYUd decreases from 0.53 to 0.40 with a linear head, while DINOv3's mIoU on ADE20K increases from 55.9% to 59.7%. These consistent gains demonstrate that language-based spatial knowledge transfer effectively enhances visual encoders' spatial understanding capabilities.

### 4.3 COMPLEX 3D-CENTRIC TASKS

**Setup.** We evaluate SpatialBoost on Lexicon3D (Man et al., 2024), a unified benchmark for 3D scene understanding covering vision-language reasoning, visual grounding, semantic understanding, and geometric understanding. Following Lexicon3D protocols, we freeze visual backbones and train task-specific heads (see Section A for details).

**Results.** As shown in Table 3, SpatialBoost shows comprehensive improvements across diverse 3D tasks. OpenCLIP's BLEU-1 improves from 36.9 to 39.2 on ScanQA (Azuma et al., 2022), while DINOv3 increases from 51.4 to 54.9 on SQA3D (Ma et al., 2023), demonstrating that SpatialBoost improves spatial understanding without compromising language capabilities. Notably, SigLIPv2's 3D semantic segmentation dramatically improves from 6.9 to 54.9 mIoU, highlighting SpatialBoost can inject robust spatial knowledge into encoders with initially limited spatial awareness.

### 4.4 VISION-BASED ROBOT LEARNING

**Setup.** We evaluate SpatialBoost on vision-based robot control using 4 domains from CortexBench (Majumdar et al., 2023) spanning locomotion and manipulation tasks (Rajeswaran et al., 2017; Yu et al., 2020; Tassa et al., 2018; Wüthrich et al., 2020). Following CortexBench protocols, we train behavior cloning agents using [CLS] representations to predict expert actions from visual observations. We report the mean of best performance across 5 evaluation runs (see Section A for details).

Table 5: **Results on image classification and retrieval tasks.** We report Top-1 accuracy of kNN performance and linear probing (lin.) for image classification on validation set of ImageNet-1K (Russakovsky et al., 2015). For image retrieval, we report global average precision (GAP) on Met (Ypsilantis et al., 2021) and mean average precision (mAP) on Oxford-Hard (Oxford-H) (Radenović et al., 2018), Paris-Hard (Paris-H) (Radenović et al., 2018), and AmsterTime dataset (Yildiz et al., 2022). For all results, we freeze the encoder backbone.

| | Image classification | | Image retrieval | | | |
|---|---|---|---|---|---|---|
| Method | ImageNet (kNN) | ImageNet (lin.) | Oxford-H | Paris-H | Met (GAP) | AmsterTime |
| OpenCLIP | 84.0 | 86.8 | 23.4 | 59.7 | 7.4 | 24.4 |
| +SpatialBoost (Ours) | **86.1** | **87.9** | **32.8** | **69.4** | **19.7** | **30.3** |
| SigLIPv2 | 86.3 | 89.1 | 25.1 | 60.9 | 13.9 | 15.5 |
| +SpatialBoost (Ours) | **87.6** | **90.0** | **36.0** | **69.1** | **24.0** | **27.2** |
| DINOv2 | 84.5 | 87.3 | 58.2 | 84.6 | 44.6 | 48.9 |
| +SpatialBoost (Ours) | **86.4** | **88.6** | **61.3** | **85.2** | **45.1** | **50.8** |
| DINOv3 | 85.8 | 88.4 | 60.7 | 87.1 | 55.4 | 56.5 |
| +SpatialBoost (Ours) | **87.7** | **90.2** | **64.1** | **88.6** | **57.0** | **56.9** |

(a) SigLIPv2 depth estimation    (b) DINOv3 depth estimation    (c) Semantic segmentation

Figure 5: **Effect of dataset scalability.** We investigate the effect of the size of analysis of data scalability effects on (a) depth estimation results (AbsRel, RMSE) on NYUd benchmark for SigLIPv2, (b) depth estimation results (AbsRel, RMSE) on NYUd benchmark for DINOv3, and (c) semantic segmentation results (mIoU) on ADE20K benchmark for SigLIPv2 and DINOv3. The results show scalable performance improvements with increased data size.

**Results.** As shown in Table 4, SpatialBoost significantly improves robot task performance across all vision encoders. For example, DINOv2 + SpatialBoost achieves 68.1% on Adroit versus 55.4% for DINOv2 alone, demonstrating that enhanced spatial representations directly benefit robot control.

### 4.5 IMAGE CLASSIFICATION AND RETRIEVAL TASKS

**Setup.** We evaluate SpatialBoost's impact on instance recognition using ImageNet-1K (Russakovsky et al., 2015) classification and retrieval benchmarks (Oxford, Paris (Radenović et al., 2018), Met (Ypsilantis et al., 2021), AmsterTime (Yildiz et al., 2022)). Following DINOv3 protocols, we use linear probing on [CLS] representations for classification and similarity-based ranking for retrieval (see Section A for details).

**Results.** As shown in Table 5, SpatialBoost improves both classification and retrieval despite these tasks not explicitly requiring spatial understanding. DINOv3's ImageNet accuracy increases from 88.4% to 90.2%, while Oxford-Hard mAP improves from 60.7 to 64.1. These results demonstrate that SpatialBoost enhances general vision capabilities without overfitting to spatial features, likely due to our dual-channel attention preserving pre-trained knowledge and the inclusion of general scene captions alongside spatial reasoning.

### 4.6 ABLATION STUDY AND ANALYSIS

**Effect of LLM-based Fine-tuning.** In Table 6, we investigate whether LLM-based decoders provide superior supervision compared to pixel-level alternatives. We fine-tune the vision encoder with linear layer, SAM (Kirillov et al., 2023) decoder, VGGT (Wang et al., 2025a) decoder, and LLM (Yang et al., 2024). We then evaluate encoders on ImageNet-1K classification, ADE20K segmentation, and NYUd depth estimation. The results show that LLM consistently outperform pixel-level supervision methods, validating that language provides superior dense information transfer for vision encoders (see Section E for details).

Table 6: **Effect of LLM-based fine-tuning.** We fine-tune the vision encoder with different headers. We report accuracy (%) for classification (Cls) on ImageNet-1K, mIoU for segmentation (Seg) on ADE20K, RMSE for depth estimation on NYUd, and BLEU-1 score for vision-language reasoning (VLR) on ScanQA. We use ViT-L/14 as the backbone architecture of the encoder.

| Method | Cls ↑ | Seg ↑ | Depth ↓ | VLR ↑ |
|---|---|---|---|---|
| DINOv2 | 86.3 | 47.7 | 0.38 | 39.2 |
| +Linear (depth) | 85.7 (-1.39%) | 47.9 (+0.42%) | 0.35 (-7.89%) | 36.9 (-5.87%) |
| +Linear (seg.) | 86.6 (+0.35%) | 48.8 (+2.31%) | 0.45 (+18.42%) | 37.1 (-5.36%) |
| +SAM decoder | 86.3 (+0.0%) | 50.1 (+5.03%) | 0.42 (+10.53%) | 37.6 (-4.08%) |
| +VGGT decoder | 84.8 (-1.74%) | 45.6 (-4.40%) | 0.35 (-7.89%) | 37.3 (-4.85%) |
| +LLM (Ours) | 88.3 (+2.32%) | 51.5 (+7.97%) | 0.32 (-15.79%) | 40.0 (+2.04%) |

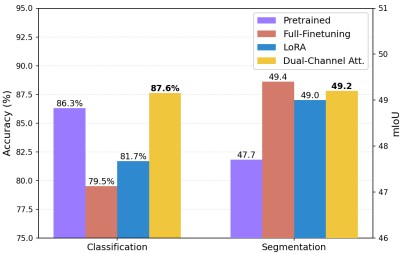

Figure 6: **Effect of dual-channel attention layer.** We report the linear evaluation performance of DINOv2-ViT-L/14 across different fine-tuning strategies.

Table 7: **Component-wise analysis.** We investigate the effect of multi-turn spatial reasoning data and the effect of single-view and multi-view data. Multi-turn order means the order of three levels (*i.e.*, pixel, object, and scene) in our visual spatial reasoning data.

| Method | Multi-turn order | Single-view data | Multi-view data | Cls ↑ | Seg ↑ | Depth ↓ |
|---|---|---|---|---|---|---|
| DINOv2 | ✗ | - | - | 86.3 | 47.7 | 0.38 |
| +SpatialBoost | Reverse | +100K | - | 87.4 | 48.4 | 0.35 |
| | Random | +100K | - | 87.4 | 48.5 | 0.36 |
| | Forward | +100K | - | 87.6 | 48.9 | 0.34 |
| | Forward | - | +100K | 87.6 | 48.2 | 0.36 |
| | Forward | +50K | +50K | 87.6 | 49.2 | 0.32 |

**Effect of Multi-turn Visual Reasoning.** In Table 7, we investigate how the hierarchical structure of reasoning affects representation learning. We compare dataset construction strategies: (a) shuffled multi-turn, (b) reversed order (scene→object→pixel), and (c) forward order (pixel→object→scene). The forward hierarchical ordering shows optimal performance, demonstrating that reasoning order significantly impacts the quality of representation.

**Effect of Single-view and Multi-view Data.** In Table 7, we investigate the effect of single-view and multi-view reasoning data. With fixed total samples, we compare single-view only, multi-view only, and combined training. While both data types independently improve performance, the combination achieves the highest results, confirming their complementary nature.

**Comparison with Naive Post-training.** In Table 8, we investigate the effect of post-training. With fixed total samples (*i.e.*, 300K data in multi-turn reasoning data), we compare the naive post-training scheme and SpatialBoost. We evaluate the performance of the vision encoder across five tasks: depth estimation, segmentation, vision-language reasoning, robot learning, and classification. The results show that naive post-training does not yield effective representations for downstream tasks.

**Effect of Dual-channel Attention Layer.** In Figure 6, we investigate whether our dual-channel attention mechanism preserves pre-trained knowledge during fine-tuning. We evaluate several approaches for fine-tuning the vision encoder including full fine-tuning, LoRA (Hu et al., 2021), and dual-channel (Hong et al., 2023a) on ImageNet (Russakovsky et al., 2015) and ADE20K (Zhou et al., 2017). Dual-channel attention uniquely preserves and even enhances pre-trained knowledge, while other approaches cause degradation.

**Dataset Scalability.** We analyze the impact of dataset sizes on depth estimation results from NYUd (Silberman et al., 2012) benchmark and semantic segmentation results from ADE20K (Zhou et al., 2017) benchmark. With matched training iterations (*i.e.*, one epoch for 300K data), larger datasets yield consistent improvements, indicating robust scalability potential.

## 5  CONCLUSION

In this paper, we have presented SpatialBoost, a framework to enhance the vision encoders by leveraging linguistic expressions of geometric and semantic information within images. SpatialBoost

Table 8: **Effect of post-training.** We fine-tune vision encoders with their original pre-training objectives (simple FT). We report RMSE for monocular depth estimation on NYUd, mIoU for semantic segmentation on ADE20K, BLEU-1 score for vision-language reasoning on ScanQA, average score for robot learning on CortexBench, and Top-1 accuracy (%) for classification on ImageNet-1K.

| Method | Depth Estimation ↓ | Segmentation ↑ | Vision-Language Reasoning ↑ | Robot Learning ↑ | Classification ↑ |
|---|---|---|---|---|---|
| OpenCLIP | 0.53 | 39.5 | 36.9 | 65.5 | 84.0 |
| +Simple FT | 0.56 | 39.6 | 37.7 | 63.7 | 84.3 |
| +SpatialBoost (Ours) | **0.40** | **40.5** | **39.2** | **72.9** | **86.1** |
| SigLIPv2 | 0.51 | 42.8 | 38.1 | 69.7 | 86.3 |
| +Simple FT | 0.53 | 43.0 | 38.4 | 67.9 | 86.4 |
| +SpatialBoost (Ours) | **0.39** | **45.1** | **40.8** | **75.8** | **87.6** |
| DINOv2 | 0.37 | 49.3 | 39.5 | 68.1 | 84.5 |
| +Simple FT | 0.36 | 49.6 | 39.4 | 69.4 | 84.7 |
| +SpatialBoost (Ours) | **0.30** | **52.0** | **40.3** | **75.8** | **86.4** |
| DINOv3 | 0.31 | 55.9 | 40.6 | 72.8 | 85.8 |
| +Simple FT | 0.31 | 56.4 | 40.2 | 75.5 | 86.1 |
| +SpatialBoost (Ours) | **0.25** | **59.7** | **43.3** | **80.8** | **87.7** |

uses LLM and dual-channel attention layers to exploit linguistic information into image representations, generates a multi-turn visual spatial reasoning dataset, and leverages them to improve the image representations. Our experiments show that SpatialBoost consistently enhances the vision encoders on various downstream tasks that require a spatial understanding of images. We hope that our work further facilitates future research on designing and enhancing vision encoders.

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

# A IMPLEMENTATION DETAILS

## A.1 TRAINING DETAILS OF STAGE 1 & 2

We train our multi-modal architecture with 4x NVIDIA Tesla A100s. In multi-modal architecture, we choose Qwen-2.0-7B (Yang et al., 2024) as the LLM backbone and 2-layer MLP as the projector. In feature alignment pre-training (Stage 1), we train the projector on a BLIP-558K data (Liu et al., 2024a) for one epoch with a learning rate of 2e-3 and a batch size of 256. In visual instruction tuning (Stage 2), we fine-tune both the projector and the LLM backbone on the LLaVA-Instruct-158K dataset (Liu et al., 2024a) and our multi-view VQA dataset (described in Section 3.2) for one epoch with a batch size of 128.

## A.2 TRAINING DETAILS OF STAGE 3

In this stage, we adapt dual-channel attention layers in training vision encoders by introducing additional attention channels described in Section 3.1. By applying dual-channel attention, the number of model parameters increased by 30% in OpenCLIP and SigLIPv2 and by 25% in DINOv2 and DINOv3, respectively. We freeze the LLM decoder and fine-tune the vision encoder and projector on a multi-turn visual spatial reasoning dataset (described in Section 3.2) for one epoch with a learning rate of 2e-5 and a batch size of 128. We conduct hyperparameter search for the learning rate from 1e-6 to 1e-2.

## A.3 DENSE PREDICTION TASKS

From the vision encoder obtained through SpatialBoost, we performed depth estimation and semantic segmentation. We follow the same protocol as in DINOv2 (Oquab et al., 2023), defining three primary hyperparameters for our linear probing setup: the learning rate, the number of output layers, and whether we concatenate the average-pooled patch token features with the class token. Concretely, we perform a grid search over learning rates in 1e-4 to 1e-1, choose the output layers from $\{1, 4\}$, and optionally concatenate average-pooled representations. We train each linear layer with SGD for 12500 iterations using random-resized-crop data augmentation. We then select the best hyperparameter combination on validation accuracy.

## A.4 3D SCENE UNDERSTANDING

We evaluate whether SpatialBoost enables complex 3D-centric reasoning using the Lexicon3D (Man et al., 2024) benchmark. Lexicon3D provides a unified probing framework that freezes visual backbones and attaches task-specific heads to evaluate vision-language reasoning, visual grounding, 3D semantic segmentation, and geometric correspondence. Following the Lexicon3D protocol, we extract features from 2D vision encoders and evaluate them on various 3D understanding tasks.

**Vision-Language Reasoning.** To evaluate vision-language reasoning, we target the 3D visual question-answering (3D-VQA) on ScanQA (Azuma et al., 2022) and SQA3D (Ma et al., 2023) datasets. We follow the 3D-LLM (Hong et al., 2023b) architecture as our task head. Specifically, we use a Q-Former module (Li et al., 2023a) to project multi-view visual features into the input space of the language model. These projected features are then fed to the LLM (*e.g.*, FlanT5 (Chung et al., 2024)) for generating answers. We pre-train only the Q-Former projection module for 10 epochs using 3D-Language dataset (Hong et al., 2023b) and fine-tune the module for 35 epochs using training split of ScanQA and SQA3D. We keep both the vision encoder and LLM frozen during training.

**Visual Grounding.** To evaluate visual grounding with vision encoder, we target the object localization task based on text descriptions on the ScanRefer (Chen et al., 2020a) dataset. We use an attention-based fusion head following Multi3DRefer (Zhang et al., 2023). The task head consists of a multi-layer attention module with 4 transformer layers that fuses visual and text embeddings. After projecting multi-view features to 3D space and extracting object features via average pooling within ground-truth bounding boxes, we apply cross-attention between object features and CLIP-encoded text descriptions. The fusion module outputs confidence scores for each object. We train the header for 30 epochs with cross-entropy loss.

**Geometric Understanding.** To evaluate geometric understanding, we target the geometric correspondence task. We adopt a REGTR-style (Yew & Lee, 2022) transformer cross-encoder as the task head. The head process features from two partial point clouds to establish correspondences. After obtaining point correspondences through the transformer, we apply the Kabsch-Umeyama (Kabsch, 1976; Umeyama, 2002) algorithm for closed-form estimation of rotation and translation parameters. We train the transformer head using partial scene registration benchmark (Man et al., 2024) for 30 epochs using a combination of correspondence loss and transformation loss.

**3D Semantic Understanding.** To evaluate 3D semantic understanding, we target the point-wise semantic classification task on ScanNet (Dai et al., 2017). We employ a linear probing head consisting of a single fully-connected layer followed by sigmoid activation: $\mathbf{y} = \text{Sigmoid}(\text{FC}(\mathbf{x}))$, where $\mathbf{x} \in \mathbb{R}^{N \times d}$ represents projected point features from multi-view images, $\mathbf{y} \in \mathbb{R}^{N \times C}$ represents class probabilities for $C = 20$ semantic classes and $N$ is the number of points in each point cloud. The linear layer maps from feature dimension $d$ to the number of classes. We train the linear layer using ScanNet segmentation dataset with cross-entropy loss at learning rate 1e-4 for 20 epochs.

## A.5 VISION-BASED ROBOT LEARNING

We train the robot agents using 100 demos for each task. For training, we use keypoint augmentation (James & Davison, 2022) for each demonstration, and use the end-effector controller with path planning as an action mode. We use the front camera of 224×224 resolution without depth measurements. We evaluate the model 5 times by training with a pre-defined interval and report the mean of the best performance.

## A.6 IMAGE CLASSIFICATION TASK

We train a linear classifier on top of the `[CLS]` token from the last feature of the vision encoder using the training split of ImageNet-1K (Deng et al., 2009) dataset. Following the evaluation protocol of DINOv3 (Siméoni et al., 2025), we employ SGD optimizer with momentum 0.9 and random-resized-crop data augmentation. We train the linear layer for 10 epochs with a batch size of 1024. We perform a grid search for the optimal learning rate, ranging from 1e-4 to 1e-1, selecting the best performing configuration.

## A.7 IMAGE RETRIEVAL TASK

We evaluate the image retrieval performance of vision encoders using a non-parametric retrieval approach. Specifically, we compute cosine similarity between the output `[CLS]` tokens of query and target images to establish ranking. For Oxford (Radenović et al., 2018), Paris (Radenović et al., 2018), and AmsterTime (Yildiz et al., 2022) datasets, we resize images to $224 \times 224$ resolution, while for the Met (Ypsilantis et al., 2021) dataset, we resize to the nearest multiple of the patch size. All other setups follow evaluation protocols of each benchmark.

# B  ADDITIONAL EXPERIMENTAL RESULTS

## B.1  VISUAL QUESTION-ANSWERING (VQA) TASKS

Table 9: Effect of the vision encoder on spatial reasoning and general VQA benchmarks.

| Model | Vision encoder | Spatial Reasoning | | General VQA | | | |
|---|---|---|---|---|---|---|---|
| | | SpatialRGPT | BLINK-D | VQAv2 | GQA | SQA-I | MME |
| GPT-4o | - | 39.7 | 72.6 | - | - | - | - |
| Gemini-2.5-Flash | - | 42.5 | 77.4 | - | - | - | - |
| Vicuna-1.5-7B | OpenCLIP | 13.3 | 51.6 | 78.5 | 62.0 | 66.8 | 1510.7 |
| | +SpatialBoost | **52.0** | **84.9** | **79.0** | **65.6** | **67.1** | **1516.3** |
| | SigLIPv2 | 21.1 | 52.3 | 79.4 | 62.5 | 66.8 | 1519.4 |
| | +SpatialBoost | **61.3** | **87.5** | **80.0** | **69.1** | **69.5** | **1527.6** |
| | DINOv2 | 18.8 | 55.2 | 75.2 | 61.5 | 66.0 | 1509.2 |
| | +SpatialBoost | **54.2** | **87.2** | **76.8** | **62.5** | **66.8** | **1514.2** |
| | DINOv3 | 17.6 | 53.9 | 78.7 | 61.9 | 65.8 | 1514.7 |
| | +SpatialBoost | **58.7** | **87.9** | **80.0** | **65.5** | **67.1** | **1520.6** |

**Setup.** To investigate whether SpatialBoost can enhance visual representations by capturing geometric and semantic information within images, we evaluate our framework on VQA tasks that require (1) 3D geometric spatial reasoning and (2) general knowledge.

For spatial reasoning, we consider the VQA tasks from SpatialRGPT-bench (Cheng et al., 2024) and BLINK's Relative Depth Benchmark (*i.e.*, BLINK-D) (Fu et al., 2024b), where the goal is to predict the relative or absolute positional relations between objects. For general VQA, we consider widely-used benchmarks such as VQAv2 (Goyal et al., 2017), GQA (Hudson & Manning, 2019), SQA-I (Lu et al., 2022), and MME (Fu et al., 2024a). Given our SpatialBoost vision encoders, we follow the setup in LLaVA-1.5 (Liu et al., 2024a) that trains the LLM backbone (Vicuna-1.5-7B (Zheng et al., 2023)) and the 2-layer MLP projector in two stages while freezing our vision encoder.

**Details for Spatial Reasoning.** The SpatialRGPT-Benchmark is designed to assess 3D spatial understanding across a diverse range of scenes, incorporating both quantitative and qualitative QAs. We evaluate BLINK's Relative Depth Benchmark for depth comparison between the coordinates of two objects. Given that these benchmarks allow for multiple correct answers, leveraging an LLM-based evaluation provides a reasonable and consistent approach to judging model responses. For this, we utilize GPT-4 (Achiam et al., 2023) to determine the accuracy of the responses. For qualitative questions, responses are assessed on 0 to 1 scoring scale. For quantitative questions, the LLM extracts numerical values from answers and model responses and standardizes them to a same unit for comparison. We use judging prompts following SpatialRGPT (Cheng et al., 2024).

In Table 9, we use the closed-source large vision language models (LVLMs), although they are not directly compared to our approach. We provide the versions of the closed-source LVLMs as follows:

- `openai/gpt-4o-2024-11-20`

- `Google/gemini-2.5-flash-preview-04-17`

**Results.** As shown in Table 9, we observe that SpatialBoost consistently and significantly enhances both the spatial reasoning capabilities and general knowledge of existing vision-language models, even though only the frozen vision encoder was changed. For instance, Vicuna-1.5-7B with SpatialBoost DINOv3 raises the score 17.6 to 58.7 on SpatialRGPT benchmark, surpassing frontier models like GPT-4o (Achiam et al., 2023) (39.7) and Gemini-2.5-Flash (DeepMind, 2025) (42.5). This demonstrates that our framework can indeed induce representations that are useful for solving complex QA tasks that require spatial understanding while preserving or even improving its general knowledge.

## C Multi-view VQA Dataset

We utilize multi-view data to inject rich 3D information into vision encoders. We found that proper instruction tuning is crucial for LLMs to stably transfer the 3D information to vision encoders. However, existing datasets are limited to enhance multi-view understanding, as most VQA datasets focus exclusively on single-view scenarios. We thereby construct a multi-view VQA dataset.

We consider both 3D datasets and ego-centric video data for our multi-view VQA construction. Specifically, we utilize ScanNet (Dai et al., 2017), Mip-NeRF360 (Barron et al., 2022), and MVImgNet (Yu et al., 2023) for 3D data, and Ego4D (Grauman et al., 2022) for ego-centric video data. From these datasets, we extract pairs of images that satisfy the following LPIPS (Zhang et al., 2018) constraint:

$$0.35 \leq \text{LPIPS}(\mathbf{x_i}, \mathbf{x_j}) \leq 0.65, \text{ where } \mathbf{x_i}, \mathbf{x_j} \in \{\mathbf{x_1} \cdots \mathbf{x_N}\}. \quad (2)$$

This constraint effectively filters out outlier samples for meaningful multi-view learning. Given the selected image pairs, we utilize GPT-4o (Achiam et al., 2023) to generate three types of visual questions: (1) common VQA, (2) adversarial VQA, and (3) multi-choice VQA. These question types are designed to probe general knowledge understanding from multi-view visual inputs, thereby guiding the model to accurately process and answer multi-view visual questions. We provide specific prompts used for generating multi-view VQA data in Table 10.

Table 10: Prompt examples for generating multi-view VQA data.

```
system_prompt =[
  "You are a helpful multimodal assistant.
  Generate question-answer pairs for given two images.
  Both images are came from same scene.
  When referring to the image, please call it the first image or the second image."
]
general_vqa_prompt =[
  "Please give me an exact question and answer by referring to the images.
  This is a common VQA.
  Create relevant question about these 2 images,
  referencing details that may only be visible if we consider both views.
  Then provide a concise, correct answer.
  The answer should be in length between 10 and 80 words."
]
multi_choice_vqa_prompt =[
  "Please give me an exact question and answer by referring to the images.
  This is a multi-choice VQA.
  Create relevant question about these 2 images,
  referencing details that may only be visible if we consider both views.
  Then also generate 4 answer candidates,
  where only one candidate is correct and the others are very wrong.
  List candidates A to D or 1 to 4.
  The answer is the index of correct question.
  Each candidates should be in length between 5 and 20 words.
  ]
```

# D    MULTI-TURN VISUAL SPATIAL REASONING DATASET

We here provide a detailed implementation of the data generation pipeline and examples of multi-turn visual spatial reasoning.

We construct a multi-turn visual spatial reasoning dataset by associating each single-view image $\mathbf{x}$ or multi-view images $\{\mathbf{x}_1 \cdots \mathbf{x}_N\}$ with 12 sequential QA turns. The first 5 turns focus on pixel-level view, prompting questions about point-wise depth or depth comparisons. The next 4 turns shift to object-level queries, referring to approximate bounding cubes (*i.e.*, 3D bounding boxes) for each object. The next one turn addresses scene-level understanding, requiring holistic 3D interpretation. The last 2 turns are GPT-generated scene captions for given image input. For instance, the entire sequence of question-answer pairs for image $\mathbf{x}$ is described by

$$\text{Pixel-level} : \left(Q_{\mathbf{x}}^{(1)}, A_{\mathbf{x}}^{(1)}\right) \to \cdots \to \left(Q_{\mathbf{x}}^{(5)}, A_{\mathbf{x}}^{(5)}\right) \to,$$
$$\text{Object-level} : \left(Q_{\mathbf{x}}^{(6)}, A_{\mathbf{x}}^{(6)}\right) \to \cdots \to \left(Q_{\mathbf{x}}^{(9)}, A_{\mathbf{x}}^{(9)}\right) \to,$$
$$\text{Scene-level} : \left(Q_{\mathbf{x}}^{(10)}, A_{\mathbf{x}}^{(10)}\right) \to,$$
$$\text{Scene Caption} : \left(Q_{\mathbf{x}}^{(11)}, A_{\mathbf{x}}^{(11)}\right) \to \left(Q_{\mathbf{x}}^{(12)}, A_{\mathbf{x}}^{(12)}\right).$$

Each turn builds on the previous answers, allowing the LLM to engage in CoT reasoning. To extract 3D information for each image, we use the specialized vision models (*e.g.*, depth and segmentation networks) and synthesize QA pairs that reflect the relevant 3D information, ensuring that the final scene-level query can integrate pixel-level and object-level details into a coherent spatial understanding.

**Filtering for Single-view Image.** Generating visual spatial reasoning data requires multiple objects in an image. Therefore, selecting the appropriate images is necessary. Following SpatialVLM (Chen et al., 2024a) and SpatialRGPT (Cheng et al., 2024), we adopt a CLIP-based open-vocabulary classification model (Sun et al., 2023) to identify appropriate images with 100K samples from 314K samples of SA1B (Kirillov et al., 2023). We provide the labels to get filtered images in Table 11.

Table 11: CLIP labels for filtering images.

| Label type | Labels |
|---|---|
| Positive labels | "an iPhone photo of an indoor scene"
"an iphone photo of an outdoor scene"
"a DSLR photo of an indoor scene"
"a DSLR of an outdoor scene" |
| Negative labels | "a close up shot of a single object"
"a product displayed in front of a white background"
"an artwork"
"a painting"
"a screenshot of a graphical user interface"
"a piece of text"
"a sketch" |

**Filtering for Multi-view Images.** We apply LPIPS (Zhang et al., 2018) metric to 3D data (*e.g.*, ScanNet (Dai et al., 2017) trainset) and ego-centric video data (*e.g.*, Ego4D (Grauman et al., 2022)) to obtain pairs of images that satisfy Equation (2). This constraint prevents sampling of image pairs that are either too dissimilar or overly redundant from the datasets.

**Point Cloud Processing.** We process two types of input: (1) single-view and (2) multi-view. For a single-view image, we use the results of the segmentation and depth estimation to generate a 3D point cloud for objects in images. In particular, we use Depth-pro (Bochkovskii et al., 2024) to perform metric depth estimation. For multi-view images, we obtain a 3D point cloud through VGGT (Wang et al., 2025a), which is a state-of-the-art 3D reconstruction model. For each image input $\{\mathbf{x}_1 \cdots \mathbf{x}_N\}$, we first select an image $\mathbf{x}_i$, where $\mathbf{x}_i \in \{\mathbf{x}_1 \cdots \mathbf{x}_N\}$, among the image input and generate pixel-level data by randomly selecting the 2D coordinates of bounding boxes in $\mathbf{x}_i$ and then extract the depth information. We also generate object and scene-level data by randomly selecting the bounding cubes obtained by using 3D point cloud. We represent the bounding cubes in the canonical space, which is proposed by SpatialVLM (Chen et al., 2024a).

Table 12: Template examples for pixel-level VQA.

```
single_point_questions =[
  "What is the depth value at pixel point [A]?"
  "How far away is point [A]?"
  "Tell me the depth of point [A]."
]
single_point_answers =[
  "[X] away."
  "It is [X]."
  "Depth value of point [A] is [X]."
]
close_predicate_questions =[
  "Which point is close to a viewer?  Point:  [A], Point:  [B]."
  "Is point [A] closer than [B]?"
  "Which point has a smaller depth value?  Point [A] or Point [B]?"
  "Compare the depth of point [A] and point [B]."
]
close_true_responses =[
  "Yes, point [A] is closer to the viewer than point [B]."
  "Indeed, point [A] has a smaller depth value than point [B]."
  "Correct, point [A] is closer than point [B]."
]
close_false_responses =[
  "No, point [A] is not closer than point [B]."
  "In fact, point [B] is closer to the viewer than point [A]."
  "Incorrect, point [B] has a smaller depth value than point [A]."
]
```

**Pixel-level VQA Data.** Pixel-level dataset has two types of QAs: (1) single point and (2) multi point QA. Single-point QA consists of questions that query depth values at specific coordinates on the image, and multi point QA involves comparing depth values between two different coordinates. To avoid generating excessively noisy data, all depth values are rounded to the third decimal place. We use a centimeter scale for depth values less than 0.5 meters while maintaining the template. We provide examples of templates for each type of QA of this level in Table 12.

Table 13: Template examples for object-level VQA.

```
bounding_cube_questions =[
  "Identify [A] and [B]"
  "What is the center of the 3d bounding box coordinate for [A]?"
]
bounding_cube_answers =[
  "[X]"
  "Center:  [X]"
  "[A] in [X] and [B] in [Y]"
]
left_predicate_questions =[
  "Is the [A] to the left of the [B] from the viewer's perspective?"
  "Does the [A] appear on the left side of the [B]?"
  "Can you confirm if the [A] is positioned to the left of the [B]?"
]
left_true_responses =[
  "Yes, the [A] is to the left of the [B]."
  "Indeed, the [A] is positioned on the left side of the [B]."
  "Correct, you'll find the [A] to the left of the [B]."
]
left_false_responses =[
  "No, the [A] is not to the left of the [B]."
  "In fact, the [A] is either to the right of or directly aligned with the [B]."
  "Incorrect, the [A] is not on the left side of the [B]."
]
```

**Object-level VQA Data.** Object-level dataset has two types of QAs: (1) predicting a bounding cube of an object from the bounding box of the object, and (2) predicting the relative positional relationship between two objects. We provide examples of templates for each type of QA of this level in Table 13.

**Scene-level VQA Data.** Scene-level dataset has single type of QA: predicting the 3D relative distance between two objects. We provide examples of templates for each type of QA of this level in Table 14.

Table 14: Template examples for scene-level VQA.

```
distance_questions =[
  "What is the distance between the [A] and the [B]?"
  "How far is the [A] from the [B]?"
  "How distant is the [A] from the [B]?"
  "Measure the distance from the [A] to the [B]."
]
distance_answers =[
  "[X]"
  "the [A] and the [B] are [X] apart."
  "They are [X] apart."
  "The distance of the [A] from the [B] is [X]."
]
```

**Expand Viewpoints in Multi-view Data.** Through the aforementioned process, we obtain multi-view reasoning data for 2-view images. We denote these obtained views as anchor views. To extend beyond 2-view configurations, we additionally sample interpolated frames between the anchor views and validate whether the VQA pairs generated for the anchor views remain valid for these new viewpoints using GPT-4o. Specifically, if the existing VQA pairs are verified as correct for more than half of the interpolated views, we incorporate these interpolated views as additional viewpoints. This approach enables us to extend the 2-view input to arbitrary multi-view configurations. Among our 200K multi-view samples, we have 160K 2-view samples, 30K 4-view samples, and 10K 8-view samples.

## E  DETAILS OF ABLATION STUDY AND ANALYSIS

We here provide a detailed implementation of ablation study and analysis.

### E.1  COMPARISON ON DIFFERENT HEADERS

Our key hypothesis is that language supervision, particularly through LLM-based supervised fine-tuning, can effectively distill rich 3D information into vision encoders. To validate this, we investigate whether LLM provides superior supervision compared to pixel-level alternatives. We align various headers with vision encoders following the SpatialBoost framework, then fine-tune the vision encoder with dual-channel attention. We evaluate each enhanced vision encoder on ImageNet-1K (Deng et al., 2009) image classification, ADE20K (Zhou et al., 2017) semantic segmentation, and NYUd (Silberman et al., 2012) monocular depth estimation. As shown in Table 6, pixel-level supervision leads to catastrophic forgetting, while language supervision preserves pre-trained knowledge. This validates our hypothesis that language serves as an effective modality for transferring dense and hierarchical spatial information.

For all experiments, we fine-tune the vision encoder with fixed 300K samples extracted from our multi-turn visual reasoning dataset, except for the VGGT experiment. We choose DINOv2-ViT-L/14 as a vision encoder architecture, with following evaluation protocols for each downstream task detailed in Section A. The specific implementation for each header-based fine-tuning approach is provided in following paragraphs:

**SAM Decoder.** We adopt the SAM decoder as a header and introduce an MLP layer to match dimensions with the vision encoder. Following the SpatialBoost training strategy, we first align only the MLP layer using 300K samples from SA1B (Kirillov et al., 2023) dataset. Subsequently, we apply dual-channel attention to the vision encoder and fine-tune it using 300K segmentation samples from our multi-turn visual reasoning dataset, which is also sampled from SA1B dataset.

**VGGT Decoder.** VGGT (Wang et al., 2025a) is a state-of-the-art 3D reconstruction model that employs DINOv2-ViT-L/14-reg (Darcet et al., 2023) as a feature extractor. Building upon this off-the-shelf pipeline, we apply dual-channel attention to the vision encoder and perform fine-tuning. We utilize 300K 3D data samples from Co3D (Reizenstein et al., 2021) for training.

**Linear Layers.** We consider two different pixel-level modalities as input for linear layers: (1) depth and (2) segmentation. As linear layers are randomly initialized, we first train the linear layer while freezing the vision encoder. We use 300K samples from SA1B to train the linear layer, then apply dual-channel attention to the vision encoder and fine-tune the vision encoder with 300K samples from our reasoning data. For depth data, we use depth maps obtained through Depth-Pro (Bochkovskii et al., 2024) on SA1B and a subset of our reasoning dataset. For segmentation data, we follow the same data configuration in SAM decoder experiment.

**LLM (Ours).** We use Qwen-2.0-7B (Yang et al., 2024) as the LLM backbone. We train the projector with 300K SA1B and fine-tune the vision encoder and projector with 300K samples from our reasoning data. We follow all other training setup described in Section A.

# F ADDITIONAL ANALYSIS

## F.1 DETAILED ANALYSIS ON REASONING HIERARCHY

In this section, we investigate which components of the multi-turn visual reasoning data contribute most significantly to the performance of SpatialBoost. We provide a detailed analysis.

Table 15: Effect of reasoning hierarchy.

| Method | Depth ↓ | | | | Segmentation ↑ | | | | Classification ↑ | |
|---|---|---|---|---|---|---|---|---|---|---|
| | OpenCLIP | | DINOv2 | | OpenCLIP | | DINOv2 | | OpenCLIP | DINOv2 |
| | lin. | DPT | lin. | DPT | lin. | +ms | lin. | +ms | | |
| Pre-trained | 0.56 | 0.41 | 0.38 | 0.29 | 39.1 | 45.7 | 47.7 | 53.1 | 83.9 | 86.3 |
| Pix | 0.52 | 0.40 | 0.34 | 0.29 | 39.6 | 46.3 | 48.2 | 53.4 | 84.0 | 86.6 |
| Obj | 0.53 | 0.41 | 0.37 | 0.30 | 39.4 | 46.3 | 48.0 | 53.3 | 84.7 | 87.2 |
| Scene | 0.53 | 0.41 | 0.38 | 0.32 | 39.2 | 45.9 | 47.7 | 53.3 | 84.5 | 87.1 |
| Pix + Obj | 0.44 | 0.39 | 0.35 | 0.28 | 39.8 | 46.6 | 48.8 | 53.5 | 84.7 | 87.3 |
| Pix + Scene | 0.46 | 0.40 | 0.36 | 0.28 | 39.5 | 46.5 | 48.5 | 53.4 | 84.4 | 87.2 |
| Obj + Scene | 0.51 | 0.42 | 0.39 | 0.31 | 39.5 | 46.5 | 47.6 | 53.3 | 85.0 | 87.4 |
| Pix + Obj + Scene | **0.42** | **0.39** | **0.32** | **0.27** | **40.0** | **46.9** | **49.2** | **54.2** | **85.1** | **87.6** |

**Setup.** We explore which levels of the reasoning hierarchy have an impact on the performance of SpatialBoost by measuring the performance of vision encoders fine-tuned with different combinations of reasoning levels. For all experiments, we fix the sample size at 100K and ensure identical ratio for each combination. We evaluate monocular depth estimation on NYUd (RMSE), semantic segmentation on ADE20K (mIoU), and classification on ImageNet-1K (Top-1 accuracy). We use ViT-L/14 as a vision encoder architecture in all experiments. All other setups are the same as described in Section A.

**Results.** As shown in Table 15, we observe that pixel-level QA and its combinations remark superior performance in dense prediction tasks, indicating pixel-level QA aids in higher-level understanding. We also observe that object-level QA and its combinations achieve strong improvements in classification. The results highlights that the combination with all levels achieves the best performance across all tasks, validating the effectiveness of our hierarchical reasoning.

## F.2 DETAILED ANALYSIS ON SINGLE-VIEW AND MULTI-VIEW DATA

In this section, we investigate the effect of single-view and multi-view data on various downstream tasks. We provide a detailed analysis.

Table 16: Effect of single-view and multi-view data across diverse tasks.

| Model | SV | MV | Cls ↑ | Seg ↑ | Depth ↓ | VLR | | VG | GU | | 3D SU | |
|---|---|---|---|---|---|---|---|---|---|---|---|---|
| | | | | | | ScanQA ↑ | SQA3D ↑ | ScanRef ↑ | RR@0.05m ↑ | RTE ↓ | Acc ↑ | mIoU ↑ |
| SigLIPv2 | - | - | 89.1 | 42.8 | 0.51 | 38.1 | 48.5 | 51.4 | 47.8 | 0.28 | 47.7 | 9.2 |
| | +200K | +100K | 90.2 | 44.7 | 0.41 | 40.5 | 50.0 | 56.6 | 84.1 | 0.18 | 77.7 | 51.8 |
| | +150K | +150K | 90.0 | 44.9 | 0.39 | 40.6 | 50.1 | 56.6 | 84.9 | 0.16 | 80.2 | 52.4 |
| | +100K | +200K | 90.0 | **45.1** | **0.39** | **40.8** | **50.1** | **56.8** | **86.4** | **0.15** | **81.0** | **55.5** |
| DINOv3 | - | - | 88.4 | 55.9 | 0.31 | 40.6 | 51.4 | 56.2 | 86.9 | 0.10 | 91.1 | 69.1 |
| | +200K | +100K | 90.2 | 59.5 | 0.27 | 43.1 | 54.7 | 61.1 | 96.0 | 0.08 | 91.4 | 69.7 |
| | +150K | +150K | **90.3** | 59.6 | 0.26 | 43.1 | **55.0** | 61.1 | 96.9 | 0.07 | 91.6 | 70.2 |
| | +100K | +200K | 90.2 | **59.7** | **0.25** | **43.3** | 54.9 | **61.1** | **97.5** | **0.06** | **91.9** | 70.6 |

**Setup.** We explore the effect of single-view and multi-view data by fine-tuning the vision encoder with different proportions of our reasoning data. With fixed total samples, *i.e.*, 300K from multi-turn spatial reasoning data, we train the vision encoders and evaluate them on classification (Cls) on ImageNet-1K, segmentation on ADE20K, depth estimation on NYUd, and 3D-centric tasks on Lexicon3D (Man et al., 2024). We use ViT-g/16 and ViT-7B/16 as the architecture of SigLIPv2 and DINOv3, respectively. All other setups are the same as described in Section A.

**Results.** As shown in Table 16, we observe that multi-view reasoning data leads to improvements in tasks which require spatial knowledge such as depth estimation, segmentation, geometric under-

standing (GU), and 3D semantic understanding (3D SU). Following the size of multi-view data, SigLIPv2's GU registration recall improves from 84.1% to 86.4%, and 3D SU mIoU improves from 51.8% to 55.5%. These results demonstrate that multi-view reasoning data can effectively enhance 3D understanding of the vision encoder.

### F.3 DETAILED ANALYSIS ON DUAL-CHANNEL ATTENTION

We provide quantitative and qualitative results for dual-channel attention (see Table 17 and Figure 7).

Table 17: Quantitative results of dual-channel attention.

| Method | Classification ↑ | Segmentation ↑ | Depth estimation ↓ |
|---|---|---|---|
| DINOv2 (Pre-trained) | 86.3 | 47.7 | 0.38 |
| Full Fine-tuning | 79.5 | 49.4 | 0.31 |
| LoRA | 81.7 | 49.0 | 0.32 |
| Dual-Channel Attn. | 87.6 | 49.2 | 0.32 |

**Setup.** We evaluate different fine-tuning methodologies while fixing the reasoning data sample size at 100K. Performance is measured on ImageNet-1K classification (accuracy), ADE20K segmentation (mIoU), and NYUd depth estimation (RMSE). All experiments utilize DINOv2 with ViT-L/14 architecture.

**Results.** As shown in Table 17, we find that full fine-tuning and LoRA similarly exhibits performance drops in classification. In contrast, dual-channel attention shows consistent performance improvements across all tasks. This indicates that dual-channel attention effectively enhances spatial capabilities while preventing overfitting to spatial-specific features, maintaining the generalization ability. Partial results of Table 17 are visualized in Figure 6.

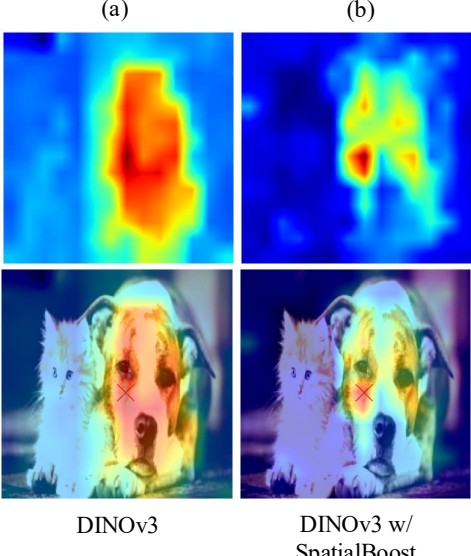

Figure 7: **Qualitative results for dual-channel attention.** We visualize attention heatmap from (a) DINOv3 and (b) SpatialBoost DINOv3. We rollout attention layers for cosine similarity between patches. Red cross denotes a query patch. We visualize pure attention heatmap (top) and RGB overlayed version (bottom).

### F.4 DETAILED RESULTS ON DATA SCALABILITY

We provide more detailed results for data scalability. In Table 18, SpatialBoost improves SigLIPv2 and DINOv3 in all tasks.

Table 18: Data scalability on classification, segmentation, and depth estimation.

| Model | Sample size | Classification ↑ | Segmentation ↑ | Depth estimation ↓ |
|---|---|---|---|---|
| SigLIPv2 | Pre-trained | 89.1 | 42.8 | 0.51 |
| +SpatialBoost | 50K | 89.5 | 43.2 | 0.44 |
| | 100K | 89.7 | 44.5 | 0.42 |
| | 300K | 90.0 | 45.1 | 0.39 |
| DINOv3 | Pre-trained | 88.4 | 55.9 | 0.31 |
| +SpatialBoost | 50K | 88.6 | 56.8 | 0.29 |
| | 100K | 90.0 | 58.3 | 0.28 |
| | 300K | 90.2 | 59.7 | 0.25 |

## F.5 ANALYSIS ON BIAS PROPAGATION IN REASONING DATA

We provide an analysis of bias in vision foundation models used to generate spatial reasoning data.

Table 19: Comparison between VFM-based and GT-based reasoning data.

| Method | Cls ↑ | Seg ↑ | Depth ↓ | VLR ↑ |
|---|---|---|---|---|
| DINOv2 | 86.3 | 47.7 | 0.38 | 39.2 |
| +VFM-based | 87.5 | 48.7 | 0.34 | 39.6 |
| +GT-based | 87.5 | 48.8 | 0.34 | 36.9 |
| Δ (VFM − GT) | 0.0 | -0.1 | 0.0 | 0.0 |

**Setup.** We explore the effect of bias propagation from vision foundation models (*e.g.*, SAM, Depth-pro) used to generate spatial reasoning data. With fixed 100K ScanNet (Dai et al., 2017) single-view samples, we generate reasoning data based on 3D metadata extracted from vision foundation models (VFM-based) and ScanNet ground-truth annotation (GT-based). We then fine-tune the vision encoder and evaluate the performance on ImageNet-1K classification (Cls), ADE20K segmentation, NYUd depth estimation, and ScanQA vision-language reasoning (VLR).

**Results.** As shown in Table 19, we observe that the performance between VFM-based and GT-based is negligible. The results demonstrate that the effect of bias propagation is marginal in our reasoning data pipeline.

## F.6 ADDITIONAL RESULTS ON MULTI-MODAL LARGE LANGUAGE MODELS

We provide results of application our framework on Multi-modal Large Language Models (MLLM).

Table 20: Effect of SpatialBoost on MLLM visual encoders.

| Method | #Params | Cls ↑ | Seg ↑ | Depth ↓ |
|---|---|---|---|---|
| InternViT-6B-v2.5 | 5.5B | 86.6 | 39.4 | 0.46 |
| +SpatialBoost (Ours) | 6.0B | **89.1** | **48.5** | **0.35** |
| Qwen3-VL-VE | 0.6B | 87.9 | 40.8 | 0.44 |
| +SpatialBoost (Ours) | 0.7B | **89.3** | **44.3** | **0.36** |

**Setup.** We apply SpatialBoost on the vision encoders of InternVL-3 (Zhu et al., 2025) and Qwen3-VL (Yang et al., 2025). With fixed 300K samples from our reasoning data, we fine-tune the vision encoder and evaluate linear probing for ImageNet-1K classification, ADE20K segmentation, and NYUd depth estimation.

Additionally, we evaluate the performance of MLLM with SpatialBoost encoder on VQA tasks targeting multi-modal reasoning (MMMU (Yue et al., 2024)), real world comprehension (Real-WorldQA (xAI org., 2024)), OCR and document understanding (OCRBench (Liu et al., 2024b), DocVQA (Mathew et al., 2021)), multi-image comprehension (BLINK (Fu et al., 2024b), MUIR-Bench (Wang et al., 2024a)), and embodied reasoning (ERQA (Abeyruwan et al., 2025)).

Table 21: Effect of SpatialBoost on MLLM VQA performance.

| Method | MMMU | RealWorldQA | OCRBench | DocVQA | BLINK | MUIRBench | ERQA |
|---|---|---|---|---|---|---|---|
| InternVL 3-38B | 70.1 | 75.6 | 886 | 95.4 | 64.0 | 63.8 | 42.8 |
| +SpatialBoost (Ours) | **70.8** | **75.9** | **894** | 95.4 | **69.2** | **70.7** | **49.3** |
| Qwen3-VL-32B-Instruct | 76.0 | 79.0 | 895 | 96.9 | 67.3 | 72.8 | 48.8 |
| +SpatialBoost (Ours) | **76.4** | **79.6** | **909** | **97.1** | **70.8** | **76.4** | **51.5** |

**Results.** As shown in Table 20, we observe that SpatialBoost produces notable performance gain in the vision encoders of Qwen3-VL and InternVL3. For example, InternViT-6B-v2.5 with Spatial-Boost raises the mIoU 39.4 to 48.5 on segmentation task. In Table 21, we observe that SpatialBoost yield consistent performance improvements on diverse VQA tasks. For instance, Qwen3-VL with SpatialBoost vision encoder rises the score from 72.8 to 76.4 on MUIRBench and from 48.8 to 51.5 on ERQA.

## G USE OF AI TOOLS

We acknowledge that a large language model (LLM) was used to refine the phrasing and grammar of the manuscript.

