# OpenReview forum: "SpatialBoost: Enhancing Visual Representation through Language-Guided Reasoning"
_ICLR.cc/2026/Conference — Submitted to ICLR 2026_

### Official Review · Reviewer_sYwN · 2025-10-31

**Soundness:** 4
**Presentation:** 3
**Contribution:** 3
**Rating:** 6
**Confidence:** 3

**Summary:**

This paper proposes SpatialBoost, a novel framework for enhancing the spatial awareness of pre-trained vision encoders. The authors identify that existing encoders, trained predominantly on 2D images, fail to capture 3D spatial relationships. Their core idea is to distill dense 3D spatial knowledge, derived from specialist vision models (e.g., for depth estimation), into linguistic form and then use this language-based supervision to fine-tune the encoder. The framework involves two key stages: 1) generating a multi-turn, hierarchical Chain-of-Thought (CoT) dataset where a Large Language Model (LLM) converts geometric information into a series of structured questions and answers, progressing from pixel-level to scene-level understanding; and 2) fine-tuning the vision encoder using this dataset. To prevent catastrophic forgetting of the encoder's original capabilities, they employ a dual-channel attention mechanism that adds trainable parameters alongside the frozen original ones. The authors demonstrate that applying SpatialBoost to state-of-the-art encoders like DINOv3 and SigLIPv2 leads to consistent performance improvements across an extensive and diverse set of downstream tasks, including dense prediction, 3D scene understanding, robotic control, and even general image classification and retrieval.

**Strengths:**

1. **Novel and Effective Framework:** The central idea of using language as an intermediate representation to "teach" a vision encoder about 3D space is both clever and highly effective. Instead of relying on complex architectural changes or end-to-end training with limited 3D data, the framework elegantly distills knowledge from powerful specialist models into a format (structured language) that can be easily injected into the encoder via an LLM-based training loop.
2. **Comprehensive and Rigorous Evaluation:** The experimental validation is exceptionally strong and is a major highlight of the paper. The authors evaluate their method across a very wide range of challenging benchmarks:
    - **Geometric/Dense Tasks:** Monocular depth estimation (NYUd, KITTI) and semantic segmentation (ADE20K, Pascal VOC).
    - **Complex 3D Tasks:** The Lexicon3D benchmark, covering 3D VQA, visual grounding, and geometric understanding.
    - **Downstream Applications:** Vision-based robot learning on CortexBench.
    - **General Vision Tasks:** ImageNet classification and multiple image retrieval benchmarks.
    The fact that SpatialBoost provides consistent, and often significant, improvements across all these diverse domains is very convincing.
3. **Successfully Addresses Catastrophic Forgetting:** A critical challenge when fine-tuning foundation models is preventing the degradation of their general capabilities. The paper's use of a dual-channel attention mechanism is a simple yet powerful solution. The ablation in Figure 6 and the results in Table 5 are particularly compelling, showing that not only does SpatialBoost avoid performance loss on ImageNet classification, it actually *improves* upon the original SoTA encoders. This is a very strong result, suggesting the injected spatial knowledge also acts as a useful regularizer for general vision tasks.
4. **High-Quality Data Generation Process:** The design of the multi-turn, hierarchical (pixel → object → scene) VQA dataset is well-reasoned. The ablation study in Table 7 confirms that this structured, forward-reasoning approach is more effective than random or reversed orders, validating the CoT-inspired design.

**Weaknesses:**

1.  **Complexity and Cost of Data Pipeline:** The data generation process, while effective, is quite complex and relies on a cascade of pre-existing models (depth estimators, segmentation models, 3D reconstruction models) as well as a powerful proprietary LLM (GPT-4o). This makes the framework potentially expensive and difficult to replicate or extend to new domains without significant engineering effort.

2.  **Dependence on External Models for Ground Truth:** The quality of the supervision signal is fundamentally capped by the performance of the specialist models used to extract 3D information and the LLM used to convert it to text. Any errors or biases from these external models are directly propagated into the training data for the vision encoder.

3.  **Clarity in Presentation:** While generally well-written, some sections could benefit from greater clarity. The initial overview in Figure 1 is a bit high-level, and a more detailed architectural diagram combining the data generation and training loops would be helpful for understanding the full system at a glance.

**Questions:**

See Weaknesses

---

> ### Author Response · Authors · 2025-11-22
> **Response to Reviewer sYwN**
>
> Dear Reviewer sYwN,
>
> We sincerely appreciate your insightful comments and effort in reviewing our manuscript. We address each of your concerns below, with key revisions highlighted in “blue”.
>
> ---
>
> **[W1] Complexity and cost of the data pipeline**
>
> **[A1]** We respectfully argue that our pipeline is not complex in the conventional sense. Our spatial reasoning data pipeline does not rely on difficult rule-based heuristics, nor is it model-dependent. The process requires only a single forward pass through lightweight vision models with minimal computational cost. Subsequently, we generate CoT data using a template-based approach from the extracted 3D metadata, and this core generation process does not involve GPT-4o at all. GPT-4o is used only for two auxiliary purposes: (1) VQA verifier when optionally extending 2-view data to N(>2)-view configurations, (2) generating a general description per image.
>
> In addition, the modular design of our pipeline enables straightforward enhancement through plug-and-play integration of advanced vision foundation models. Furthermore, our reasoning data pipeline is designed to target general images and can accommodate diverse domains (e.g., indoor and outdoor scenes) without requiring domain-specific adjustments. Moreover, to facilitate research fields, we plan to release our source code.
>
> ---
>
> **[W2] Dependency on external models and potential bias propagation**
>
> **[A2]** We acknowledge your concern that biases from vision foundation models (VFM) may propagate to the vision encoder. However, we observe that across diverse downstream tasks, SpatialBoost consistently achieves performance improvements without degradation in any task, suggesting that bias issues are marginal in practice. Nevertheless, to rigorously examine the impact of potential biases, we conducted controlled experiments comparing VFM-based supervision against Ground-Truth-based supervision using 100K single-view samples from ScanNet.
>
> As shown in the table below, we find that even when using ground-truth annotations, the vision encoder performance shows negligible differences compared to VFM-based generation. This marginal difference provides empirical evidence that VFM biases do not significantly affect encoder training under our framework, validating the robustness of our data pipeline. We incorporated these updated experimental results into the revised manuscript (see Table 19 and Appendix F.5).
>
> \begin{array}{lcccc}
> \hline
> \text{Method} & \text{Cls}\uparrow & \text{Seg}\uparrow & \text{Depth}\downarrow & \text{VLR}\uparrow \newline
> \hline
> \text{DINOv2 ViT-L/14} & 86.3 & 47.7 & 0.38 & 39.2 \newline
> \text{+ VFM-based} & 87.5 & 48.7 & 0.34 & 39.6 \newline
> \text{+ GT-based} & 87.5 & 48.8 & 0.34 & 39.6 \newline
> \hline
> \Delta \text{(VFM - GT)} & 0.0 & -0.1 & 0.0 & 0.0 \newline
> \hline
> \end{array}
>
> ---
>
> **[W3] Clarity in Figure 1**
>
> **[A3]** We appreciate your constructive feedback. Following your suggestions, we revised Figure 1 to provide a more comprehensive and integrated overview that encompasses the data generation pipeline, the architecture, and the model training. We believe this enhanced figure improves clarity and enables readers to grasp the full system more readily.

---

> > ### Comment · Reviewer_sYwN · 2025-11-27
> >
> > Dear authors
> >
> > Thank you for the detailed responses, which address most of my concerns. I am inclined to keep my current score of acceptance.

---

> ### Author Response · Authors · 2025-11-26
> **Further Discussion Before the Deadline**
>
> Dear Reviewer sYwN,
>
>
> We sincerely appreciate your time and thoughtful efforts in reviewing our manuscript.
>
>
> As the discussion period is nearing its end within a week, we would like to gently remind you in case you have any remaining comments. We believe that we have sincerely and successfully addressed your concerns, supported by the corresponding additional experimental results.
>
>
> If you have any questions or concerns, please don't hesitate to let us know. We remain fully available for further discussion and are prepared to conduct additional experiments to address any remaining concerns during the discussion period.
>
>
> Best regards,\
> Authors

---

### Official Review · Reviewer_cDb7 · 2025-10-31

**Soundness:** 3
**Presentation:** 3
**Contribution:** 3
**Rating:** 8
**Confidence:** 4

**Summary:**

This paper presents SpatialBoost, a technique that inject spatial knowledge into the representations of visual foundation models through LLMs. Specifically, it finetunes the representations from visual foundation models by building up a LLaVA-like architecture to project visual foundation model features to the token space of LLMs. Afterwards, the whole pipeline is tuned with spatial-related tasks like multi-view VQA and multi-turn visual reasoning. Experimental results demonstrate that after finetuning the visual foundation models with SpatialBoost, the features achieve better performance for a variety of tasks including dense predictions, 3D understanding, robot learning, and high-level tasks like image classification and retrieval.

**Strengths:**

- The proposed method of using LLM supervision to enhance the spatial information for visual representations is simple but effective. The motivation is reasonable, and the solution is natural and intuitive.

- After applying the feature finetuning method SpatialBoost, the visual representations have universal improved performance on various types of tasks including dense predictions, 3D understanding, robot learning, and even high-level tasks like image classification and retrieval. It is very challenging from my opinion, and hard to achieve by previous feature finetuning approaches, because many finetuning methods on visual representations could actually result in a performance tradeoff across different types of tasks, which is also demonstrated in Table 6 in the paper. Therefore, I think the proposed SpatialBoost is a well-rounded solution for enhancing the visual representations.

- The proposed SpatialBoost can bring performance gain by injecting spatial information to many kinds of foundation model features, including CLIP, SigLIP, DINOv2, and even the most recent DINOv3, which demonstrates the robustness, generalizability, and broad application of this approach.

- The presentation of this paper is smooth and clear, which makes the proposed solution easy to understand.

**Weaknesses:**

- I think it is mostly a well-written paper with effective solutions and comprehensive experimental evaluations. One thing that I find a bit confusing is the categorization on the multi-turn spatial reasoning task. In Figure 2, there is a pyramid showing three levels of spatial knowledge being scene-level QA, object-level QA, and pixel-level QA. However, I do not quite agree that the corresponding examples on the right (the three QA examples in Figure 2 corresponds to the left using the color match, if I understand correctly) can be categorized into these different levels of QA tasks. For example, the one shown in the blue bounding box is categorized into scene-level QA. However, it needs the model to have pixel-level localization capability to understand the spatial information of the specific pixels in the images. Therefore, I do not think the of these QA samples are quite properly categorized.

**Questions:**

- We have seen that the models finetuned on multi-view VQA and multi-turn visual spatial reasoning tasks can have universal benefit on all types of tasks. It could be interesting to further find out which downstream tasks get more benefits from which finetuning tasks. Having this information can be provide guidance for users that want to tune the visual representations for some specific downstream tasks.

- For multi-turn visual spatial reasoning, is it always following the order of scene-level QA, object-level QA, then pixel-level QA, basically to narrow down the scope of the questions? Or the questions are just sampled from all categories with random order?

---

> ### Author Response · Authors · 2025-11-22
> **Response to Reviewer cDb7 (1/3)**
>
> Dear Reviewer cDb7,
>
> We sincerely appreciate your insightful comments and effort in reviewing our manuscript. We address each of your concerns below, with key revisions highlighted in “blue”.
>
> ---
>
> **[W1] Categorization of multi-turn spatial reasoning in Figure 2**
>
> **[A1]** We appreciate your careful attention to detail. In the revision, we fix Figure 2 such that each color (blue, red, and yellow) correctly corresponds to pixel-level, object-level, and scene-level QA pairs, respectively.

---

> ### Author Response · Authors · 2025-11-22
> **Response to Reviewer cDb7 (2/3)**
>
> **[Q1] Which downstream tasks benefit most from fine-tuning tasks?**
>
> **[A2]** Following your constructive feedback, we conduct an in-depth analysis.
>
> First, to examine how single-view and multi-view data affect various downstream tasks, we fine-tune the vision encoders using 300K samples from our reasoning data with different proportions. In the table below, we observe that multi-view reasoning data leads to improvements in tasks that require spatial knowledge, such as depth estimation, segmentation, geometric understanding (GU), and 3D semantic understanding (3D SU). Following the size of multi-view data, SigLIPv2's GU registration recall improves from 84.1% to 86.4%, and 3D SU mIoU improves from 51.8% to 55.5%. These results demonstrate that multi-view data can effectively enhance 3D understanding of the vision encoder. We incorporated the analysis into the revised manuscript (see Table 16 and Appendix F.2).
>
> \begin{array}{lcc|c|c|c|cc|c|cc|cc}
> \hline
> &&&&&& \text{VLR} && \text{VG} & \text{GU} & & \text{3D SU} \newline
> \hline
> \text{Method} & \text{Single-view} & \text{Multi-view} & \text{Cls}\uparrow & \text{Seg}\uparrow & \text{Depth}\downarrow & \text{ScanQA}\uparrow & \text{SQA3D}\uparrow & \text{ScanRef}\uparrow & \text{RR\@0.05m}\uparrow & \text{RTE}\downarrow & \text{Acc}\uparrow & \text{mIoU}\uparrow \newline
> \hline
> \text{SigLIPv2} & - & - & 89.1 & 42.8 & 0.51 & 38.1 & 48.5 & 51.4 & 47.8 & 0.28 & 47.7 & 9.2 \newline
> & \text{+200K} & \text{+100K} & \bf{90.2} & 44.7 & 0.41 & 40.5 & 50.0 & 56.6 & 84.1 & 0.18 & 77.7 & 51.8 \newline
> & \text{+150K} & \text{+150K} & 90.0 & 44.9 & \bf{0.39} & 40.6 & \bf{50.1} & 56.6 & 84.9 & 0.16 & 80.2 & 52.4 \newline
> & \text{+100K} & \text{+200K} & 90.0 & \bf{45.1} & \bf{0.39} & \bf{40.8} & \bf{50.1} & \bf{56.8} & \bf{86.4} & \bf{0.15} & \bf{81.0} & \bf{55.5} \newline
> \hline
> \text{DINOv3} & - & - & 88.4 & 55.9 & 0.31 & 40.6 & 51.4 & 56.2 & 86.9 & 0.10 & 91.1 & 69.1 \newline
> & \text{+200K} & \text{+100K} & 90.2 & 59.5 & 0.27 & 43.1 & 54.7 & \bf{61.1} & 96.0 & 0.08 & 91.4 & 69.7 \newline
> & \text{+150K} & \text{+150K} & \bf{90.3} & 59.6 & 0.26 & 43.1 & \bf{55.0} & \bf{61.1} & 96.9 & 0.07 & 91.6 & 70.2 \newline
> & \text{+100K} & \text{+200K} & 90.2 & \bf{59.7} & \bf{0.25} & \bf{43.3} & 54.9 & \bf{61.1} & \bf{97.5} & \bf{0.06} & \bf{91.9} & \bf{70.6} \newline
> \hline
> \end{array}
>
> Second, to examine which granularity of data (pixel, object, scene-level) provides the benefits for each downstream task, we fine-tune the vision encoders using 100K samples in our reasoning dataset with different combinations of reasoning levels. In the table below, we observe that pixel-level data and its combinations remark superior performance in dense prediction tasks such as depth estimation and segmentation. On the other hand, we observe that object-level data and its combinations achieve strong improvements in classification. The results also highlight that the combination of all three levels (denoted as “Pix + Obj + Scene”) achieves the best performance across all tasks, validating the effectiveness of our hierarchical reasoning. We incorporated these updated experimental results into the revised manuscript (see Table 15 and Appendix F.1).
>
> \begin{array}{l|cc|cc|cc|cc|cc}
> \hline
> & \text{Depth\downarrow} & & \text{Depth\downarrow} & & \text{Segmentation}\uparrow & & \text{Segmentation}\uparrow & & \text{Classification}\uparrow \newline
> \hline
> Method & \text{OpenCLIP} & & \text{DINOv2} & & \text{OpenCLIP} & & \text{DINOv2} & &  \text{OpenCLIP} & \text{DINOv2} \newline
> \hline
>      & \text{lin.} & \text{DPT} & \text{lin.} & \text{DPT} & \text{lin.} & \text{+ms} & \text{lin.} & \text{+ms} \newline
> \hline
> \text{Pre-trained} & 0.56 & 0.41 & 0.38 & 0.29 & 39.1 & 45.7 & 47.7 & 53.1 & 83.9 & 86.3 \newline
> \hline
> \text{Pix} & 0.52 & 0.40 & 0.34 & 0.29 & 39.6 & 46.3 & 48.2 & 53.4 & 84.0 & 86.6 \newline
> \text{Obj} & 0.53 & 0.41 & 0.37 & 0.30 & 39.4 & 46.3 & 48.0 & 53.3 & 84.7 & 87.2 \newline
> \text{Scene} & 0.53 & 0.41 & 0.38 & 0.32 & 39.2 & 45.9 & 47.7 & 53.3 & 84.5 & 87.1 \newline
> \hline
> \text{Pix + Obj} & 0.44 & 0.39 & 0.35 & 0.28 & 39.8 & 46.6 & 48.8 & 53.5 & 84.7 & 87.3 \newline
> \text{Pix + Scene} & 0.46 & 0.40 & 0.36 & 0.28 & 39.5 & 46.5 & 48.5 & 53.4 & 84.4 & 87.2 \newline
> \text{Obj + Scene} & 0.51 & 0.42 & 0.39 & 0.31 & 39.5 & 46.5 & 47.6 & 53.3 & 85.0 & 87.4 \newline
> \hline
> \text{Pix + Obj + Scene} & \bf{0.42} & \bf{0.39} & \bf{0.32} & \bf{0.27}  & \bf{40.0} & \bf{46.9} & \bf{49.2} & \bf{54.2} & \bf{85.1} & \bf{87.6} \newline
> \hline
> \end{array}

---

> ### Author Response · Authors · 2025-11-22
> **Response to Reviewer cDb7 (3/3)**
>
> **[Q2] The order of multi-turn visual spatial reasoning**
>
> **[A3]** The multi-turn visual spatial reasoning data always follow the hierarchical order: pixel-level $\rightarrow$ object-level $\rightarrow$ scene-level, progressing from narrow to broad spatial understanding. This hierarchical construction is more effective than random ordering, as empirically validated in the ablation study comparing different multi-turn orders (see Table 7 Random vs. Forward).

---

> ### Author Response · Authors · 2025-11-26
> **Further Discussion Before the Deadline**
>
> Dear Reviewer cDb7,
>
>
> We sincerely appreciate your time and thoughtful efforts in reviewing our manuscript.
>
>
> As the discussion period is nearing its end within a week, we would like to gently remind you in case you have any remaining comments. We believe that we have sincerely and successfully addressed your concerns, supported by the corresponding additional experimental results.
>
>
> If you have any questions or concerns, please don't hesitate to let us know. We remain fully available for further discussion and are prepared to conduct additional experiments to address any remaining concerns during the discussion period.
>
>
> Best regards,\
> Authors

---

> > ### Comment · Reviewer_cDb7 · 2025-11-26
> >
> > Dear authors,
> >
> > Thank you for providing the response! For the in-depth analysis to Q1 and explanation of the questions orders to Q2, I think my questions are well addressed. However, for W1, after re-examine the figure, I still feel that I cannot distinguish between the illustrated object-level QA and scene-level QA. Both questions are asking about something related to "the man wearing glasses" and "the man wearing orange vest", what makes them to be categorized in object-level QA and scene-level QA respectively?

---

> > > ### Author Response · Authors · 2025-12-03
> > > **Response to Reviewer cDb7**
> > >
> > > Thank you for your continued engagement in discussion.
> > >
> > > We categorize questions that require broader scene understanding beyond individual objects as scene-level QA, which represents a higher level of reasoning complexity than object-level QA.
> > >
> > > Specifically, object-level QA can often be addressed through 2D image understanding, such as identifying single object properties or estimating relative positions between objects, which are perceivable from a single 2D observation. In contrast, scene-level QA requires 3D spatial understanding to answer questions about absolute distances or metric measurements that cannot be reliably inferred from 2D cues alone in some cases.
> > >
> > > To accurately answer questions about distance between two objects, the model should implicitly understand object-level QA which consists of (1) the perception of both objects and (2) understanding their relative spatial relationships. Furthermore, measuring the exact distance between objects requires comprehensive 3D scene understanding and spatial reconstruction. In particular, in multi-view scenarios, it requires complex scene-level knowledge integration across multiple input views as objects may appear in different views or be partially occluded.
> > >
> > > In essence, object-level QA focuses on properties and relationships perceivable from 2D observations, whereas scene-level QA demands holistic 3D spatial reasoning that integrates multiple elements of the scene. We hope this clarification addresses your concern about the categorization.

---

### Official Review · Reviewer_3LpB · 2025-11-01

**Soundness:** 2
**Presentation:** 2
**Contribution:** 2
**Rating:** 2
**Confidence:** 4

**Summary:**

This paper proposes SpatialBoost, a framework designed to enhance existing vision encoders with 3D spatial knowledge. Specifically, the method converts 3D spatial information into linguistic expressions and injects this knowledge into vision encoders through a language-guided reasoning process, utilizing a multi-turn CoT approach and a dual-channel attention mechanism to preserve pre-trained capabilities.

**Strengths:**

The work is motivated by a well-identified gap in 2D vision encoders—their inherent lack of 3D spatial understanding. The proposed approach of converting 3D information into linguistic expressions is innovative.

**Weaknesses:**

- The spatial CoT data is generated using GPT-4o. There lacks of evaluation regarding the quality of this synthetic data. GPT-4o may suffer from significant hallucinations on spatial-related questions, which could compromise data reliability.

- Experiments are conducted only on vision encoders like DINOv2, OpenCLIP, and the Qwen2.0-7B LLM. The method lacks comparison with recent state-of-the-art MLLMs such as Qwen2.5-VL or InternVL3. It remains unclear whether the approach would be effective when applied to these more advanced models.

- The use of CoT data format increases the sequence length during inference. It is important to quantify how many additional tokens this introduces and whether it leads to significantly higher computational costs.

**Questions:**

- How was the quality of the GPT-4o generated spatial CoT data assessed? What measures were taken to identify and mitigate potential hallucinations in the synthetic spatial QA pairs?

- How many additional tokens does the multi-turn CoT reasoning introduce during inference compared to standard VQA? Could you provide an analysis of the increase in computational cost?

---

> ### Author Response · Authors · 2025-11-22
> **Response to Reviewer 3LpB**
>
> Dear Reviewer 3LpB,
>
> We sincerely appreciate your insightful comments and effort in reviewing our manuscript. We address each of your concerns below, with key revisions highlighted in “blue”.
>
> ---
>
> **[W1&Q1] Quality of GPT-4o generated spatial CoT data and potential hallucinations**
>
> **[A1]** We would like to clarify that our spatial CoT data is generated using a template-based approach (see Tables 12-14) from 3D metadata extracted by vision foundation models (e.g., SAM, Depth-Pro). Instead, we only use GPT-4o to (1) generate general description about the images, which does not provide any spatial knowledge and (2) verify the extension from 2-view to N (>2) view configurations. Therefore, the effect of GPT-4o’s spatial hallucinations is nearly marginal in our spatial CoT data.
>
> ---
>
> **[W2] Application on recent SOTA MLLMs**
>
> **[A2]** We clarify that our experiments focused on vision encoders (DINOv2, OpenCLIP) and we even conduct experiments on recent SOTA encoders (DINOv3, SigLIPv2) in our manuscript. Nevertheless, to address your concern about whether SpatialBoost can be effectively applied on SOTA MLLM architectures, we additionally conducted experiments applying our framework on vision encoders of InternVL3 and Qwen3-VL, and evaluate the performance of their vision encoders. We incorporated the results into the revised manuscript (see Table 20-21 and Appendix F.6).
>
> First, we report an average accuracy on ImageNet classification (Cls), mIoU on ADE20K segmentation (Seg), and RMSE on depth estimation of the vision encoders. We observe that SpatialBoost produces notable performance gain in the vision encoders of Qwen3 and InternVL3.
>
> \begin{array}{lcccc}
> \hline
> \text{Method} & \\# \text{Params} & \text{Cls}\uparrow & \text{Seg}\uparrow & \text{Depth}\downarrow \newline
> \hline
> \text{InternViT-6B-v2.5} & \text{5.5B} & 86.6 & 39.4 & 0.46 \newline
> \text{+ SpatialBoost (Ours)} & \text{6.0B} & \bf{89.1} & \bf{48.5} & \bf{0.35} \newline
> \hline
> \text{Qwen3-VL-VE} & \text{0.6B} & 87.9 & 40.8 & 0.44 \newline
> \text{+ SpatialBoost (Ours)} & \text{0.7B} & \bf{89.3} & \bf{44.3} & \bf{0.36} \newline
> \hline
> \end{array}
>
> Second, we further evaluate the VQA performance of Qwen3-VL and InternVL3 with enhanced vision encoders by SpatialBoost. As shown in the table below, we find that SpatialBoost shows consistent performance improvements on diverse VQA tasks, demonstrating that SpatialBoost can be broadly applied across diverse MLLM architectures.
>
> \begin{array}{lccccccc}
> \hline
> \text{Method} & \text{MMMU} & \text{RealWorldQA} & \text{OCRBench} & \text{DocVQA} & \text{BLINK} & \text{MUIRBench} & \text{ERQA} \newline
> \hline
> \text{InternVL3-38B} & 70.1 & 75.6 & 886 & 95.4 & 64.0 & 63.8 & 42.8 \newline
> \text{+ SpatialBoost (Ours)} & \bf{70.8} & \bf{75.9} & \bf{894} & 95.4 & \bf{69.2} & \bf{70.7} & \bf{49.3} \newline
> \hline
> \text{Qwen3-VL-32B-Instruct} & 76.0 & 79.0 & 895 & 96.9 & 67.3 & 72.8 & 48.8 \newline
> \text{+ SpatialBoost (Ours)} & \bf{76.4} & \bf{79.6} & \bf{909} & \bf{97.1} & \bf{70.8} & \bf{76.4} & \bf{51.5} \newline
> \hline
> \end{array}
>
> ---
>
> **[W3&Q2] Additional tokens and computational cost during MLLM inference**
>
> **[A3]** We would like to clarify that the number of generated tokens by MLLM does not increase during the inference time. Spatial CoT data is used only for training the vision encoder (Stage 3) with the frozen LLM, and not used for MLLM training.

---

> ### Author Response · Authors · 2025-11-26
> **Further Discussion Before the Deadline**
>
> Dear Reviewer 3LpB,
>
>
> We sincerely appreciate your time and thoughtful efforts in reviewing our manuscript.
>
>
> As the discussion period is nearing its end within a week, we would like to gently remind you in case you have any remaining comments. We believe that we have sincerely and successfully addressed your concerns, supported by the corresponding additional experimental results.
>
>
> If you have any questions or concerns, please don't hesitate to let us know. We remain fully available for further discussion and are prepared to conduct additional experiments to address any remaining concerns during the discussion period.
>
>
> Best regards,\
> Authors

---

### Official Review · Reviewer_kPA7 · 2025-11-01

**Soundness:** 3
**Presentation:** 4
**Contribution:** 3
**Rating:** 4
**Confidence:** 4

**Summary:**

The paper hypothesizes that language supervision can provide a more effective way to inject 3D spatial knowledge into vision encoders than dense pixel-level tasks. To test this, the authors propose a VLM-based post-training framework that uses spatial reasoning Chain-of-Thought (CoT) data to refine pre-trained vision encoders. The idea is to convert dense 3D spatial information into linguistic descriptions and gradually teach the encoder hierarchical spatial understanding. Experiments show consistent improvements across a range of benchmarks involving both 3D perception and general vision tasks.

**Strengths:**

* The main idea is interesting and clear: using language-based supervision to improve spatial understanding in vision encoders.
* The method is easy to apply on top of existing pre-trained models, which makes it practical and useful for the community.
* The experiments are extensive, covering many benchmarks involving both 3D perception and general vision tasks.
* The results suggest that language supervision can transfer structured spatial knowledge into dense prediction tasks, which is an interesting insight.

**Weaknesses:**

* Although the hypothesis is interesting, it is not fully supported by the results. In Table 6, the LLM-based fine-tuning appears roughly on par with some dense prediction baselines, so the advantage of the language-based supervision is not clearly demonstrated.
* The comparison in Table 6 is also difficult to interpret because the baselines are not trained on the same data. A fair comparison on the exact same subset & same amount of data would make the claim stronger.
* Many performance gains in Tables 1–4 seem to come from the post-training data itself, rather than specifically from the language-based supervision. This raises the concern that the improvements may not require the proposed linguistic reasoning framework.

**Questions:**

Please see the weakness section.

---

> ### Author Response · Authors · 2025-11-22
> **Response to Reviewer kPA7 (1/2)**
>
> Dear Reviewer kPA7,
>
> We sincerely appreciate your insightful comments and effort in reviewing our manuscript. We address each of your concerns below, with key revisions highlighted in “blue”.
>
> ---
>
> **[W2] Fairness of comparison in Table 6**
>
> **[A2]** We would like to clarify that Table 6 presents a fair comparison using the same image samples and equal data quantities across all methods. Specifically, we first aligned each header or projector using 300K samples from SA-1B, then enhanced the encoder using 100K samples from our spatial CoT dataset. The only exception is VGGT, which required different image samples (Co3D) due to its multi-view input requirements, but was trained with the same amount of data.
>
> ---
>
> **[W1] Does language supervision provide clear advantages compared to other pixel-level supervision?**
>
> **[A1]** We would like to clarify that language supervision in the original Table 6 achieved clear and consistent improvements: +1.51% in classification, +2.52% in segmentation, and -10.53% (lower is better) in depth estimation. Specifically, language supervision achieves 7× better than segmentation supervision (0.23% vs. 1.51% on classification), 2× better than segmentation supervision (1.05% vs. 2.52% on segmentation), and 2× better than depth supervision (-5.26% vs. -10.53% on depth prediction).
>
> Crucially, as mentioned by Reviewer cDb7, language supervision is the only method that avoids performance drops across all tasks, which addresses a critical challenge that other supervision methods struggle to overcome. Nevertheless, to further validate the superiority of language supervision, we scaled up the encoder training data from 100K to 300K samples.
>
> As shown in the table below, language supervision is the only method achieving scalable improvements across all tasks. Notably, in the Vision-Language Reasoning (VLR) task (ScanQA BLEU-1), all pixel-level supervision methods show significant performance degradation, while our method uniquely achieves performance gains (+2.04%). Overall, the results strongly validate our hypothesis regarding the effectiveness of language supervision for transferring dense 3D spatial knowledge. We incorporated these updated experimental results into the revised manuscript (see Table 6 and Appendix E).
>
> \begin{array}{lcccc}
> \hline
> \text{Method} & \text{Cls}\uparrow & \text{Seg}\uparrow & \text{Depth}\downarrow & \text{VLR}\uparrow \newline
> \hline
> \text{DINOv2 ViT-L/14} & 86.3 & 47.7 & 0.38 & 39.2 \newline
> \text{+ Linear (depth)} & 85.1 (-1.39\\%) & 47.9 (+0.42 \\%) & 0.35 (-7.89\\%) & 36.9 (-5.87\\%) \newline
> \text{+ Linear (seg.)} & 86.6 (+0.35\\%) & 48.8 (+2.31\\%) & 0.45 (+18.42\\%) & 37.1 (-5.36\\%) \newline
> \text{+ SAM decoder} & 86.3 (+0.00\\%) & 50.1 (+5.03\\%) & 0.42 (+10.53\\%) & 37.6 (-4.08\\%) \newline
> \text{+ VGGT decoder} & 84.8 (-1.74\\%) & 45.6 (-4.40\\%) & 0.35 (-7.89\\%) & 37.3 (-4.85\\%) \newline
> \text{+ LLM decoder (Ours)} & \bf{88.3 (+2.32\\%)} & \bf{51.5 (+7.97\\%)} & \bf{0.32 (-15.79\\%)} & \bf{40.0 (+2.04\\%)} \newline
> \hline
> \end{array}

---

> ### Author Response · Authors · 2025-11-22
> **Response to Reviewer kPA7 (2/2)**
>
> **[W3] Did the improvements come from the methodology or just simply adding more data?**
>
> **[A3]** To address this concern that our gain comes from post-training rather than our language-based supervision, we compare ours with a naive post-training scheme, i.e., we fine-tuned each vision encoder using 300K samples (that were also used for training ours) with their original pre-training objectives (denoted as "Simple FT") and measured downstream performance across five tasks.
>
> As expected and presented in the table below, we observe that Simple FT exhibits performance degradation on specific tasks (e.g., SigLIPv2 on depth and robot tasks, DINOv3 on VLR) due to catastrophic forgetting, while our method consistently outperforms Simple FT across all tasks and all encoders without performance degradation. These results confirm again that simply fine-tuning off-the-shelf encoders using additional training data does not yield effective representations for downstream tasks. Taken together with the table in [W1], our findings demonstrate that LLM-based language supervision is the most effective among various explicit supervision approaches. We incorporated these updated experimental results into the revised manuscript (see Table 8).
>
> \begin{array}{lccccc}
> \hline
> \text{Method} & \text{Depth}\downarrow & \text{Seg}\uparrow & \text{VLR}\uparrow & \text{Robot}\uparrow & \text{Cls}\uparrow \newline
> \hline
> \text{OpenCLIP} & 0.53 & 39.5 & 36.9 & 65.5 & 84.0 \newline
> \text{+ Simple FT} & 0.56 & 39.6 & 37.7 & 63.7 & 84.3 \newline
> \text{\bf{+ Ours}} & \bf{0.40} & \bf{40.5} & \bf{39.2} & \bf{72.9} & \bf{86.1} \newline
> \hline
> \text{SigLIPv2} & 0.51 & 42.8 & 38.1 & 69.7 & 86.3 \newline
> \text{+ Simple FT} & 0.53 & 43.0 & 38.4 & 67.9 & 86.4 \newline
> \text{\bf{+ Ours}} & \bf{0.39} & \bf{45.1} & \bf{40.8} & \bf{75.8} & \bf{87.6} \newline
> \hline
> \text{DINOv2} & 0.37 & 49.3 & 39.5 & 68.1 & 84.5 \newline
> \text{+ Simple FT} & 0.36 & 49.6 & 39.4 & 69.4 & 84.7 \newline
> \text{\bf{+ Ours}} & \bf{0.30} & \bf{52.0} & \bf{40.3} & \bf{75.8} & \bf{86.4} \newline
> \hline
> \text{DINOv3} & 0.31 & 55.9 & 40.6 & 72.8 & 85.8 \newline
> \text{+ Simpler FT} & 0.31 & 56.4 & 40.2 & 75.5 & 86.1 \newline
> \text{\bf{+ Ours}} & \bf{0.25} & \bf{59.7} & \bf{43.3} & \bf{80.8} & \bf{87.7} \newline
> \hline
> \end{array}

---

> ### Author Response · Authors · 2025-11-26
> **Further Discussion Before the Deadline**
>
> Dear Reviewer kPA7,
>
>
> We sincerely appreciate your time and thoughtful efforts in reviewing our manuscript.
>
>
> As the discussion period is nearing its end within a week, we would like to gently remind you in case you have any remaining comments. We believe that we have sincerely and successfully addressed your concerns, supported by the corresponding additional experimental results.
>
>
> If you have any questions or concerns, please don't hesitate to let us know. We remain fully available for further discussion and are prepared to conduct additional experiments to address any remaining concerns during the discussion period.
>
>
> Best regards,\
> Authors

---

> ### Comment · Reviewer_kPA7 · 2025-11-27
>
> Thank you for the detailed response. Most of my concerns have been addressed, and I have raised my score accordingly.

---

### Author Response · Authors · 2025-11-22
**General Response**

Dear reviewers and Area Chair,

We sincerely appreciate your valuable time and effort spent reviewing our manuscript.

As reviewers highlighted, we address the important problem (3LpB, cDb7) of injecting 3D spatial knowledge into 2D vision encoders, and our method is simple yet effective (kPA7, cDb7, sYwN), and practical (kPA7). Many reviewers point out that our method is a novel (sYwN), interesting (kPA7), and innovative (3LpB) approach and can be generalizable to many foundation models (cDb7). We achieve consistent improvements across extensive benchmarks (kPA7, cDb7, sYwN), successfully address catastrophic forgetting (3LpB, sYwN) with concrete ablation studies (cDb7, sYwN).

We appreciate your constructive comments on our manuscript. In response to the comments, we have carefully revised and enhanced the manuscript with the following additional discussions and experiments:
- Comparison of language supervision and other supervision methods (Table 6, Appendix E)
- Comparison with naive post-training (Table 8)
- Detailed analysis of reasoning hierarchy (Table 15, Appendix F.1)
- Detailed analysis of single-view and multi-view reasoning data (Table 16, Appendix F.2)
- Analysis on bias propagation in reasoning data (Table 19, Appendix F.5)
- Results of SpatialBoost on Qwen3-VL and InternVL3 (Tables 20-21, Appendix F.6)
- Revision of Figure 1 for clarity
- Revision of Figure 2 for mismatched content

In the revised manuscript, we highlighted these updates in “Blue” for your convenience.

We hope our response and revision sincerely address all the reviewers’ concerns.

Thank you very much,\
Authors.

---

### Meta-Review · Area_Chair_uum8 · 2026-01-04

**Summary:**

AC ultimately agree with the negative reviewer that the core contribution does not meet the bar for acceptance, unfortunately.

The main concern is that it remains unclear whether the reported gains truly come from the proposed language-guided spatial reasoning framework, as opposed to simply adding more curated supervision and post-training data. Even after the rebuttal, the evidence does not fully disentangle the effect of the linguistic CoT formulation from the effect of additional spatial data derived from strong external models.

A second unresolved issue is the heavy dependence on external models and synthetic pipelines. Although the authors clarified that GPT-4o is not directly responsible for generating spatial facts, the supervision signal still relies on a cascade of pretrained vision models (depth, segmentation, reconstruction). This means the method largely distills existing model biases and capabilities rather than discovering new spatial representations. The rebuttal argues that bias is “marginal in practice,” but the evidence provided is limited to a small controlled setting.

The work does not provide a deeper understanding of how or when language-guided reasoning improves visual representations, nor does it establish a clear principle that would generalize beyond this specific pipeline. As a result, the contribution feels very incremental and system-driven rather than conceptually advancing representation learning.

**Reviewer Concerns:**

Well, it is still not clear what actually drives the gains. Even after the rebuttal, the results do not convincingly separate the effect of the proposed language-guided reasoning from the effect of simply adding more curated spatial supervision. The added baselines reduce some ambiguity, but they do not fully rule out that similar improvements could be obtained with alternative supervision signals that do not rely on the linguistic CoT formulation.

Further, the approach depends heavily on external models and synthetic pipelines. Although the authors clarify that GPT-4o is not directly producing spatial facts, the training signal still comes from a cascade of pretrained vision models for depth, segmentation, and reconstruction. This makes the method closer to the distillation of existing systems than learning new spatial representations, and raises concerns about bias propagation and robustness that are not fully addressed by the limited controlled experiments.

The multi-turn CoT hierarchy and dual-channel attention mainly serve to stabilize fine-tuning and avoid catastrophic forgetting. While effective in practice, they offer limited insight into when or why language-guided reasoning should improve visual representations in general IMO, and the paper lacks a principle that would extend beyond this specific VQA pipeline.

I would also note that the reliance on VQA-style benchmarks is fundamentally questionable here, since strong performance on these tasks can often be driven by language priors and superficial correlations rather than genuine improvements in spatial understanding, making it an unreliable signal for validating the core claims of the paper.

**Reviewer Scores:**

kPA7 raised to 6.

I don't think 3LpB will raise its score.

---

### Decision · Program_Chairs · 2026-01-26

Reject